# Replay Failures as Successes:
# Sample-Efficient Reinforcement Learning for Instruction Following

**Kongcheng Zhang** [1]  **Qi Yao** [2]  **Shunyu Liu** [3]  **Wenjian Zhang** [4,5]  **Min Cen** [6,7]  **Yang Zhou** [1]
**Wenkai Fang** [1]  **Yiru Zhao** [8]  **Baisheng Lai** [4,5]  **Mingli Song** [1]

## Abstract

Reinforcement Learning (RL) has shown promise for aligning Large Language Models (LLMs) to follow instructions with various constraints. Despite the encouraging results, RL improvement inevitably relies on sampling successful, high-quality responses; however, the initial model often struggles to generate responses that satisfy all constraints due to its limited capabilities, yielding sparse or indistinguishable rewards that impede learning. In this work, we propose *Hindsight instruction Replay* (HiR), a novel sample-efficient RL framework for complex instruction following tasks, which employs a *select*-then-*rewrite* strategy to *replay failed attempts as successes* based on the constraints that have been satisfied in hindsight. We perform RL on these replayed samples as well as the original ones, theoretically framing the objective as dual-preference learning at both the instruction- and response-level to enable efficient optimization using only a binary reward signal. Extensive experiments demonstrate that the proposed HiR yields promising results across different instruction following tasks, while requiring less computational budget. Our code and dataset are available at https://github.com/sastpg/HIR.

## 1. Introduction

Large Language Models (LLMs) have demonstrated remarkable capabilities across a wide spectrum of natural language tasks, such as content creation (Minaee et al., 2024; Qian et al., 2023; Lee et al., 2023), financial analysis (Arun et al., 2023; Kim et al., 2024), and robotic control (Driess et al., 2023; Firoozi et al., 2025; Huang et al., 2025a). Among these capabilities, instruction following has attracted substantial attention, driven by the growing reliance of intelligent applications on LLMs (Zhou et al., 2023a; Li et al., 2025b; Qiao et al., 2025) to reliably interpret user intent and perform specific tasks. However, real-world instructions typically involve diverse, multiple constraints, ranging from output formatting to logical consistency, which makes it challenging for LLMs to satisfy all requirements at the same time (Lior et al., 2025; Qi et al., 2025a).

Recent breakthroughs in Reinforcement Learning with Verifiable Rewards (RLVR) (Lambert et al., 2024; Guo et al., 2025; Zhang et al., 2025c) have provided a promising strategy to incentivize sophisticated reasoning patterns via rule-based rewards. Despite the leading results in mathematical analysis (Zhang et al., 2025b; Fang et al., 2025) and algorithmic programming (Zhu et al., 2025), the application of RL remains underexplored in open-ended tasks like complex instruction following (Wen et al., 2024; Sakai et al., 2025; Song et al., 2025; Zhou et al., 2025; Wang et al., 2025), where straightforward ground-truth labels are often unavailable. To bridge this gap, several recent works (Lambert et al., 2024; Peng et al., 2025; Qin et al., 2025) adopt the "LLM-as-a-Judge" paradigm, in which a powerful judge model assigns reward signals by scoring model responses against evaluable criteria derived from the instructions.

However, a critical bottleneck remains as RL relies on self-exploration to improve, yet the initial model may struggle to generate responses that satisfy all given constraints due to its limited capabilities, even after many attempts (Yue et al., 2025; Wu & Choi, 2025). As a result, the learning signal becomes highly sparse when using binary rewards (Peng et al., 2025), *i.e.*, a response is rewarded only if it perfectly meets every constraint. To mitigate this sparsity, prior works (Pyatkin et al., 2025; Qin et al., 2025) often adopt an aggregated reward signal, averaging individual scores for each constraint to provide a denser signal. Although this aggregated mechanism can stabilize training, it poses a risk of reward ambiguity. As shown in the left part of Figure 1, two

---

[1]Zhejiang University [2]Cainiao Network [3]Nanyang Technological University [4]Computer Network Information Center, Chinese Academy of Sciences [5]University of Chinese Academy of Sciences [6]University of Science and Technology of China [7]Shanghai Artificial Intelligence Laboratory [8]Alibaba Cloud Computing. Correspondence to: Shunyu Liu <shunyu.liu.cs@gmail.com>.

*Proceedings of the 43$^{rd}$ International Conference on Machine Learning*, Seoul, South Korea. PMLR 306, 2026. Copyright 2026 by the author(s).

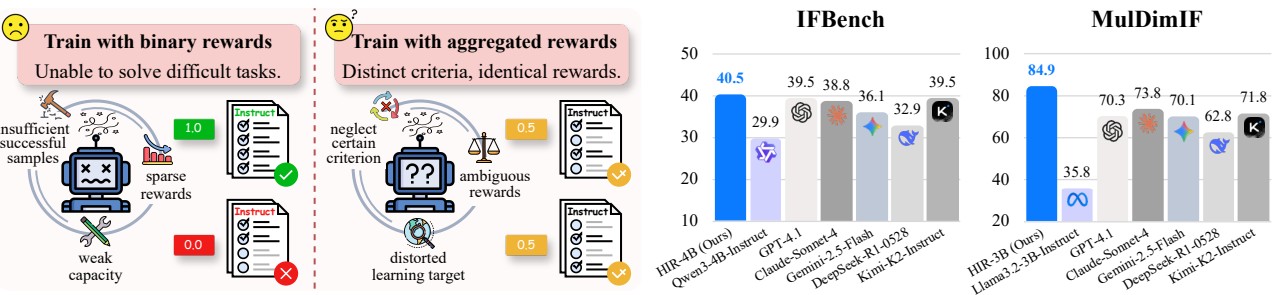

*Figure 1.* (Left) A conceptual illustration of the sparse and indistinguishable reward problem in current RLVR methods for instruction following tasks. (Right) Performance comparison between small LLMs trained by HiR and frontier LLMs on different benchmarks.

responses could share the same aggregated reward while exhibiting substantial variation in adherence to constraints, which obscures the underlying causes of failures. Worse still, this ambiguity may distort the intended learning goals: treating responses with higher rewards as preferable could misguide the model to neglect certain constraints, since both high-reward and low-reward responses may have aspects where they outperform the other.

To tackle these issues, we propose **Hindsight instruction Replay** (HiR), a sample-efficient RL framework that employs a *select*-then-*rewrite* replay strategy to solve multi-constraint instruction following tasks. Technically, we first select valuable failure samples in a curriculum-based manner, prioritizing response diversity and then gradually weighing constraint integrity as training proceeds. This trade-off dynamically accounts for the varying contribution of each sample across different learning stages, thereby improving both generalization ability and learning quality. Next, the instructions of selected samples are rewritten into "hindsight" pseudo-instructions by removing unmet constraints, followed by assigning positive rewards on these samples for replay. Finally, we perform RL on both original and replayed samples, enabling efficient learning with only a binary reward signal. The theoretical analysis reveals that our training objective not only aligns response preferences but also captures nuanced differences among instructions, facilitating the model to explicitly identify specific unmet constraints instead of relying on ambiguous rewards. Our key contributions are summarized as follows:

- We propose HiR as a novel paradigm in RL for instruction following tasks, which explores the transition of failure responses into successful ones by constructing hindsight pseudo-instructions, thereby providing more informative learning signals to enable efficient optimization.

- We introduce a *select*-then-*rewrite* replay strategy that considers both response diversity and constraint integrity, complemented by a curriculum schedule to balance the exploration-exploitation trade-off during training.

- Extensive experiments demonstrate that HiR yields results

superior to existing counterparts with even less computational budget. Notably, HiR enables small LLMs to achieve performance on par with leading LLMs, as shown in the right part of Figure 1.

## 2. Background and Notation

### 2.1. Instruction Following

Our goal is to enhance the capability of LLMs in following complex instructions. We now formally define the instruction following task. Let an instruction $q$ consist of a task description $x$ and a set of constraints $\mathcal{C} = \{c_1, c_2, \ldots, c_n\}$. Following the formulation of Zhou et al. (2023b), an LLM parameterized by $\theta$ is considered as following the instruction if its output $y$ adheres to *all* constraints in $\mathcal{C}$. We further categorize the constraints into two types inspired by Peng et al. (2025): *Hard constraints* that are verifiable via deterministic rules or code (*e.g.*, length and format); *Soft constraints* requiring semantic evaluation (*e.g.*, style or coherence). To verify whether a response meets these constraints, we adopt a hybrid evaluation approach: hard constraints are assessed using rule-based verifiers, while soft constraints are evaluated via the LLM-as-a-judge mechanism (Li et al., 2025a). The evaluation prompt for the judge LLM is presented in Appendix D. This hybrid methodology enables efficient and comprehensive evaluation of instruction adherence.

### 2.2. Evaluation Metrics

For a single constraint, we use a binary function (0 or 1) $\mathbb{I}(q, y, c_i)$ to indicate whether a response $y$ meets the specified constraint $c_i$ (true or false):

$$\mathbb{I}(q, y, c_i) = \begin{cases} \text{Rule}(c_i, y), & \text{if } c_i \in \mathcal{C}_{\text{hard}}, \\ \text{LLM}(c_i, y), & \text{if } c_i \in \mathcal{C}_{\text{soft}}, \end{cases} \quad (1)$$

where $\mathcal{C}_{\text{hard}}$ and $\mathcal{C}_{\text{soft}}$ denote the sets of hard and soft constraints, respectively. Extending this to the full constraints, we introduce two metrics at different granularities to measure performance in the following.

**Instruction-Level Accuracy (ILA).** This metric reflects

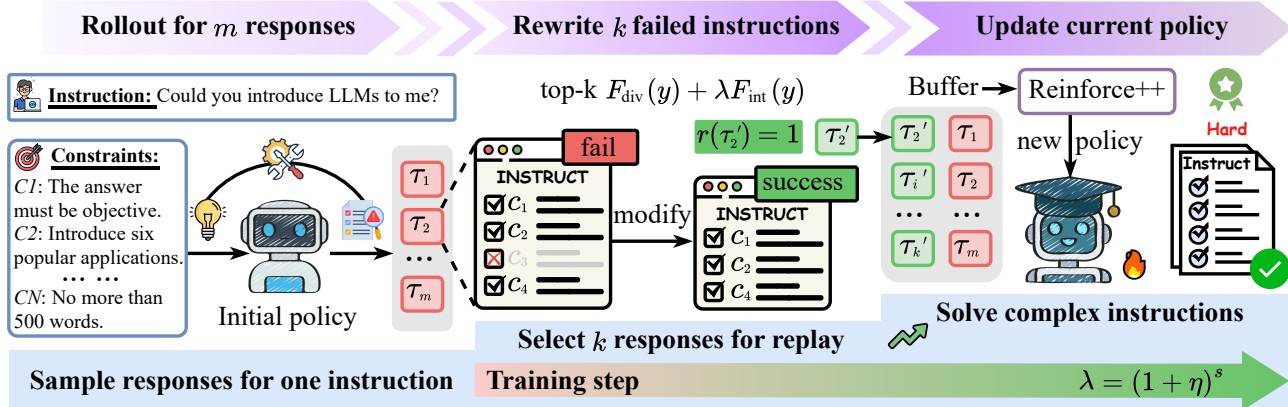

*Figure 2.* The overall framework of HiR with a *select*-then-*rewrite* replay strategy. First, we generate samples and select valuable failure attempts for replay with a curriculum schedule. Then we rewrite the instructions of selected samples into "hindsight" pseudo-instructions by removing the unmet constraints. Finally, we perform RL on both replayed samples as well as the original ones.

strict adherence to the entire instruction, where a response $y$ is considered correct only if it satisfies *every* constraint associated with the instruction $q$:

$$\text{ILA}(q, y, \mathcal{C}) = \prod_{c_i \in \mathcal{C}} \mathbb{I}(q, y, c_i). \quad (2)$$

**Constraint-Level Accuracy (CLA).** This metric measures the ability to follow individual atomic constraints, which is calculated as the percentage of satisfied constraints:

$$\text{CLA}(q, y, \mathcal{C}) = \frac{1}{|\mathcal{C}|} \sum_{c_i \in \mathcal{C}} \mathbb{I}(q, y, c_i), \quad (3)$$

When employing ILA as the reward signal for RL training, it often leads to sparse reward problem. Although CLA can provide a granular signal, it still suffers from reward ambiguity, as illustrated in the left side of Figure 1.

## 3. Hindsight Instruction Replay

During rollout generation on instructions with multiple constraints, LLMs typically fail to generate sufficient perfect responses for training, especially for models with weaker capabilities. The core idea behind our method is to learn from failures by replaying failed attempts under hindsight pseudo-instructions: despite these samples may not help models learn how to fully satisfy original instructions, they definitely tell something about how to deal with partial constraints. In what follows, we introduce our sample-efficient RL framework HiR with a *select*-then-*rewrite* replay strategy as illustrated in Figure 2.

### 3.1. *Select*-then-*Rewrite* Replay Strategy

Although replaying all partially failed attempts is possible, not all of them are equally informative to different learning stages. Samples deviating too far from the original

---

**Algorithm 1** SELECT-REWRITE($\mathcal{G}, k$)

1: **Input:** Sampling group $\mathcal{G}, k$
2: Initialize $\mathcal{T} \leftarrow \emptyset, \mathcal{H} \leftarrow \emptyset$
3: **for** each tuple $(q_i, y_i, \mathcal{C})$ in group $\mathcal{G}$ **do**
4:      Calculate score $F(y_i) = \lambda F_{int}(y_i) + F_{div}(y_i)$
5: **end for**
6: Add to $\mathcal{T}$ tuples with top-$k$ score $F(y_i)$ // *Select*
7: **for** each tuple $(q_i, y_i, \mathcal{C})$ in $\mathcal{T}$ **do**
8:      Identify satisfied constraints, *i.e.,*
     $\mathcal{C}_i' \leftarrow \{c \in \mathcal{C}_i \mid \mathbb{I}(q_i, y_i, c) = 1\}$
9:      Rewrite instruction $q_i$ as $q_i'$ using $\mathcal{C}_i'$ // *Rewrite*
10:      Add tuple $(q_i', y_i, \mathcal{C}_i')$ to $\mathcal{H}$
11: **end for**
12: **Return:** Hindsight replay buffer $\mathcal{H}$ with size $k$

---

constraints provide limited guidance toward following the targeted instructions; while some exhibit high similarity, thus redundant for learning. Consequently, different samples may contribute unevenly to the desired target. Recent studies (Hammoud et al., 2025; Xie et al., 2025) have shown that a well-designed curriculum learning approach in RL for LLMs can always improve the final performance and learning efficiency. Motivated by this, we employ a selection criterion to replay a subset of failed responses $\mathcal{T}$ from each sampling group $\mathcal{G}$ based on the scheduled *response diversity* and *constraint integrity*. Specifically, we prefer more diversity at the early training stage and gradually increase the weight on constraint integrity in our selection strategy as training proceeds, which can be formulated as the following function over the subset $\mathcal{T}$ with size $k$:

$$\mathcal{T} \triangleq \underset{\mathcal{T} \subseteq \mathcal{G}, |\mathcal{T}|=k}{\arg\max} \sum_{y \in \mathcal{T}} \big(F_{div}(y) + \lambda F_{int}(y)\big). \quad (4)$$

Under the formulation in Eq. (4), the optimal subset $\mathcal{T}$ is obtained by selecting the top-$k$ responses according to the score $F(y) = F_{div}(y) + \lambda F_{int}(y)$.

The first term $F_{div}(y)$ measures the diversity of the response. We use the response entropy to compute $F_{div}(y)$:

$$F_{div}(y) = -\frac{1}{T} \sum_{t=1}^{T} \sum_{j=1}^{V} p_{t,j} \log p_{t,j}, \quad (5)$$

where $(p_{t,1}, p_{t,2}, ..., p_{t,V}) \sim \pi_\theta(\cdot|q, y_{<t})$ denote the corresponding probability distribution of $t$-th token over model vocabulary, $V$ denotes the vocabulary size, and $T$ denotes the token length of response $y$.

The second term $F_{int}(y)$, associated with a curriculum weight $\lambda$, reflects the integrity of original constraints. It is calculated by the percentage of satisfied constraints:

$$F_{int}(y) = \frac{1}{|\mathcal{C}|} \sum_{c_i \in \mathcal{C}} \mathbb{I}(q, y, c_i). \quad (6)$$

Intuitively, the transition from response diversity to constraint integrity in our selection strategy reflects the classical exploitation–exploration trade-off. At early training stages, replaying trajectories with higher entropy encourages the model to explore uncertain yet informative patterns. However, the emphasis on diversity in later stages can distract learning, since the model has sufficiently explored the solution space and it becomes more important to focus on learning how to achieve all desired constraints of an instruction. We implement this transition by gradually increasing the weight $\lambda$ on constraint integrity during training:

$$\lambda = (1 + \eta)^s \cdot \lambda_0, \quad (7)$$

where $\eta \in [0, 1]$ is a learning pace controlling the progress of the curriculum, $s$ is the training step, and $\lambda_0$ is the initial weight for integrity. We set $\eta = 0.05$, $\lambda_0 = 2$ in this work. Intuitively, the selection process plays a role in focusing optimizations on informative failures, which improves the learning efficiency.

After selecting these responses, we rewrite their original instructions by removing the unmet constraints to construct hindsight instruction-response buffer $\mathcal{H}$, while still retaining the original pairs in the training data buffer. Specifically, the rewritten instruction $q' = x \odot c_1 \odot \cdots \odot c_j$ $(c_i \in \mathcal{C}')$, where $\odot$ denotes the string concatenation operation, and $\mathcal{C}' = \{c \in \mathcal{C} \mid \mathbb{I}(q, y, c) = 1\}$ denotes the subset of original constraints $\mathcal{C}$ that are satisfied by the response $y$. With this modification, the failed samples are assigned a non-zero rewards (set to 1 in this work) and thus facilitate learning. The *select*-then-*rewrite* process is outlined in Algorithm 1.

## 3.2. Reinforcement Learning Objective

In each sampling group $\mathcal{G}$, we first generate $m$ responses for an instruction and then select $k (k < m)$ failed responses for replay. If the number of failed responses $z$ is smaller than

$k$, we additionally generate $k - z$ supplementary samples. Finally, the model is fine-tuned on a mixed set of both the initial and replayed samples using clear binary rewards. Our HiR training objective, adapted from the Reinforce++ algorithm (Hu et al., 2025), is given by:

$$\mathcal{J}_{\text{HiR}}(\theta) = \mathbb{E}_{q \sim \mathcal{D}, \{y^{(i)}\}_{i=1}^m \sim \pi_{\text{old}}(\cdot|q), \{q'^{(i)}, y'^{(i)}\}_{i=1}^k \sim \mathcal{H}}$$

$$\underbrace{\left[ \frac{1}{m} \sum_{i=1}^{m} \frac{1}{|y^{(i)}|} \sum_{t=1}^{|y^{(i)}|} \min\left( \rho_{t,\theta}^{(i)} A_t^{(i)}, \text{clip}(\rho_{t,\theta}^{(i)}, 1 \pm \epsilon) A_t^{(i)} \right) \right.}_{\text{Objective for Initial Samples}} +$$

$$\underbrace{\frac{1}{k} \sum_{i=1}^{k} \frac{1}{|y'^{(i)}|} \sum_{t=1}^{|y'^{(i)}|} \min\left( \rho_{t,\theta}'^{(i)} A_t'^{(i)}, \text{clip}(\rho_{t,\theta}'^{(i)}, 1 \pm \epsilon) A_t'^{(i)} \right)}_{\text{Objective for Replayed Samples}} \Bigg],$$

$$(8)$$

where $\mathcal{D}$ is the dataset of instructions, $\mathcal{H}$ is the hindsight replay buffer that contains the hindsight pseudo-instruction $q'^{(i)}$ and corresponding response $y'^{(i)}$, $A_t$ denotes the advantage term for the $t$-th token in a response that is calculated based on reward. Notably, $\rho_{t,\theta}^{(i)}$ and $\rho_{t,\theta}'^{(i)}$ are the token-level importance sampling ratio between the current policy $\pi_\theta$ and old policy $\pi_{\text{old}}$:

$$\rho_{t,\theta}^{(i)} = \frac{\pi_\theta(y_t^{(i)}|q, y_{<t}^{(i)})}{\pi_{\text{old}}(y_t^{(i)}|q, y_{<t}^{(i)})}, \quad \rho_{t,\theta}'^{(i)} = \frac{\pi_\theta(y_t'^{(i)}|q'^{(i)}, y_{<t}'^{(i)})}{\pi_{\text{old}}(y_t'^{(i)}|q, y_{<t}^{(i)})}. \quad (9)$$

Note that the selection of replayed samples changes the original data distribution and therefore, the importance ratio $\rho_{t,\theta}'^{(i)}$ in Eq. (9) should not be interpreted as an unbiased correction but a PPO-inspired weighting mechanism. We further discuss this design in Appendix A.2. Algorithm 2 presents the complete HiR training procedure.

## 3.3. Theoretical Perspective

In this section, we re-examine the training objective of HiR from the perspective of preference learning. This perspective clarifies the underlying mechanism of HiR: it not only learns preference on different responses but also motivates a deeper investigation into the preference of instructions relative to a response. We first formulate the preference-based objective inspired by Rafailov et al. (2023).

**Definition 3.1.** Let $\pi_\theta$ be the language model, and $\mathcal{X}, \mathcal{Y}$ be the input and output distribution, respectively. We define the positive sample $\mathbf{y}^+ \in \mathcal{Y}$ and negative sample $\mathbf{y}^- \in \mathcal{Y}$ when $\pi_\theta$ receives a prompt $\mathbf{x} \in \mathcal{X}$, where $\mathbf{y}^+$ is preferred to $\mathbf{y}^-$ according to an underlying reward function. We define the gradient of preference learning objective as:

$$\nabla \mathcal{J}(\theta) = \mathbb{E}_{\mathbf{x}, \mathbf{y}^+, \mathbf{y}^-} [\alpha \cdot \nabla \log \pi_\theta(\mathbf{y}^+|\mathbf{x}) - \beta \cdot \nabla \log \pi_\theta(\mathbf{y}^-|\mathbf{x})]. \quad (10)$$

Here $\alpha$ and $\beta$ are both positive coefficients that weight positive and negative contributions.

**Algorithm 2** Hindsight Instruction Replay

---

**Require:** Initial policy $\pi_\theta$, Training batch data $\mathcal{D}$

1: **Input:** $m, k, \eta, \lambda_0$, reward function $r(\cdot)$
2: Initialize $\pi_{ref} \leftarrow \pi_\theta$, $\lambda \leftarrow \lambda_0$
3: **for** each training step $s$ **do**
4:    Experience buffer $\mathcal{B} \leftarrow \emptyset$
5:    **for** each $(q, \mathcal{C}) \sim \mathcal{D}$ **do**
6:       Sampling group $\mathcal{G} \leftarrow \emptyset$
7:       **for** $i = 1$ to $m$ **do**
8:          Sample response $y_i \sim \pi_\theta(\cdot|q)$
9:          Calculate reward $r_i \leftarrow \text{ILA}(q, y_i, \mathcal{C})$
10:        Store the tuple $(q, y_i, r_i)$ in buffer $\mathcal{B}, \mathcal{G}$
11:       **end for**
12:       $\mathcal{H} \leftarrow \text{SELECT-REWRITE}(\mathcal{G}, k)$
13:       **for** each tuple $(q_k, y_k, \mathcal{C}_k)$ in $\mathcal{H}$ **do**
14:          *// Replay the response under pseudo-instruction*
15:          Calculate reward $r_k \leftarrow \text{ILA}(q_k, y_k, \mathcal{C}_k)$
16:          Store the tuple $(q_k, y_k, r_k)$ in buffer $\mathcal{B}$
17:       **end for**
18:    **end for**
19:    Compute advantages $A_i$ based on rewards
20:    Update policy model $\pi_\theta$ using experience buffer $\mathcal{B}$
21:    Update $\lambda \leftarrow (1 + \eta)^s \cdot \lambda_0$
22: **end for**
23: **Return:** Trained policy $\pi_\theta$

---

**Proposition 3.2.** *The HiR objective is a form of preference learning on both the response- and instruction-level.*

$$\nabla J_{HiR}(\theta) = \mathbb{E}_{q \sim \mathcal{D}, q' \sim \mathcal{H}} \Bigg[$$

$$\underbrace{\left( \alpha_1 \mathop{\mathbb{E}}_{y^w \sim \pi_\theta(\cdot|q)} \nabla \log \pi_\theta(y^w|q) - \beta_1 \mathop{\mathbb{E}}_{y^l \sim \pi_\theta(\cdot|q)} \nabla \log \pi_\theta(y^l|q) \right)}_{\text{Response-level Preference}} +$$

$$\underbrace{\left( \alpha_2 \mathop{\mathbb{E}}_{y^r \sim \pi_\theta(\cdot|q)} \nabla \log \pi_\theta(y^r|q') - \beta_2 \mathop{\mathbb{E}}_{y^r \sim \pi_\theta(\cdot|q)} \nabla \log \pi_\theta(y^r|q) \right)}_{\text{Instruction-level Preference}} \Bigg],$$

$$(11)$$

*where $y^w$ and $y^l$ denote the winning (positive) and losing (negative) responses, $y^r$ denotes the responses that are selected for replay, $\alpha_1, \alpha_2, \beta_1, \beta_2$ are all positive values calculated based on the rewards of samples.*

**Remark.** Proposition 3.2 establishes a unified view of HiR as a dual-preference learning. While the first term aligns with standard preference on winning and losing responses, the second term introduces a discriminative signal in the instruction space. By contrasting the preference of a response under the hindsight pseudo-instruction $q'$ against the original instruction $q$, the model is encouraged to capture subtle distinctions between instructions. The detailed proof can be found in Appendix B.

# 4. Experiments

## 4.1. Experimental Setup

**Datasets and Benchmark.** To improve the capabilities of LLMs in complex instruction following tasks while balancing quantity, diversity, and quality, we collect public data from various sources, including MulDimIF (Ye et al., 2025), VerIF (Peng et al., 2025), IFTrain (Pyatkin et al., 2025), and Chatbot Arena (Zheng et al., 2023). We further synthesize constraints using programmatic approaches to enrich the dataset. After filtering and construction, we obtained the `HIR-16K` dataset, which consists of 16K queries in different scenarios, each paired with at least 5 decomposable constraints. We employ seven public benchmarks to evaluate the instruction following ability, including IFEval (Zhou et al., 2023b), IFBench (Pyatkin et al., 2025), CFBench (Zhang et al., 2025d), InfoBench (Qin et al., 2024), ComplexBench (Wen et al., 2024), MulDimIF (Ye et al., 2025) and FollowBench (Jiang et al., 2024). Additionally, we test on three out-of-domain popular reasoning benchmarks to measure its general capabilities: MATH-500 (Lightman et al., 2024), GPQA (Rein et al., 2024) and MMLU-Pro (Wang et al., 2024). Detailed dataset information is presented in Appendix C.

**Baselines and Evaluation Metrics.** We compare HiR against three categories of baselines in our experiments: (1) *SFT*: Supervised Fine Tuning on GPT-5 generated responses of our training data; (2) *DPO*: Direct Preference Optimization (Rafailov et al., 2023) and its variants RPO (Huang et al., 2025b) on pairs of chosen and rejected responses generated by GPT-5 and Qwen2.5-7B-Instruct, respectively; (3) *RL*: Reinforcement Learning with instruction-level accuracy as reward (*RL-IR*) (Peng et al., 2025) and constraint-level accuracy as reward (*RL-CR*) (Qi et al., 2025b; Pyatkin et al., 2025) on our training data. We evaluate the performance of each model by reporting its instruction-level accuracy (Eq. 2), which is the percentage of prompts that satisfy all given constraints. The specific metrics of each benchmark can be found in Appendix C.3.

**Models and Configurations.** We choose multiple models of different backbones and parameter scales for our experiments, including the Qwen2.5 series (Qwen et al., 2025) (Qwen2.5-7B-Instruct), Qwen3 series (Yang et al., 2025a) (Qwen3-4B-Instruct-2507), and Llama3.2 series (Meta, 2024) (Llama-3.2-3B-Instruct). We use verl framework (Sheng et al., 2025) to conduct RL training experiments on baselines and our method. For HiR, we sample $m = 6$ responses per prompt and replay $k = 2$ responses. For RL baselines, we sample 8 responses per prompt to ensure consistent training samples in a batch across all methods for fair comparison. We use DeepSeek-V3.1 (Liu et al., 2024a) as the judge LLM in both training and evaluation. Detailed hyperparameters are in Appendix A.1.

*Table 1.* Results on diverse instruction following dataset with different LLMs. Underline represents the best performance among all baselines, **bold** represents the best performance among all methods, and arrow indicates improvement or degradation over the initial model, and † denotes the best performance among frontier models.

| Model | IFEval | IFBench | CFBench | InfoBench | ComplexBench | MulDimIF | FollowBench |
|---|---|---|---|---|---|---|---|
| *Frontier Models* | | | | | | | |
| GPT-4.1 | 87.8 | 39.5 † | 73.2 | 60.6 † | 65.7 | 70.3 † | 86.0 † |
| DeepSeek-V3.1 | 86.1 | 34.7 | 75.6 † | 58.4 | 66.8 † | 68.3 | 83.5 |
| Gemini-2.5-Flash | 89.3 † | 36.1 | 72.8 | 57.4 | 64.4 | 70.1 | 78.5 |
| *Our Models* | | | | | | | |
| Llama-3.2-3B-Instruct | 71.2 ↑0.0 | 23.8 ↑0.0 | 31.3 ↑0.0 | 44.8 ↑0.0 | 27.6 ↑0.0 | 35.8 ↑0.0 | 58.0 ↑0.0 |
| + SFT | 73.0 ↑1.8 | 24.8 ↑1.0 | 34.6 ↑3.3 | 47.0 ↑2.2 | 26.4 ↓1.2 | 66.9 ↑31.1 | 58.1 ↑0.1 |
| + DPO | 74.3 ↑3.1 | 22.1 ↓1.7 | 40.1 ↑8.8 | 44.4 ↓0.4 | 31.2 ↑3.6 | 54.4 ↑18.6 | 61.8 ↑3.8 |
| + RPO | 75.8 ↑4.6 | 25.2 ↑1.4 | 40.3 ↑9.0 | 45.8 ↑1.0 | 30.9 ↑3.3 | 60.1 ↑24.3 | 62.2 ↑4.2 |
| + RL-IR | 77.6 ↑6.4 | 25.3 ↑1.5 | 39.2 ↑7.9 | 46.6 ↑1.8 | 29.8 ↑2.2 | 76.3 ↑40.5 | 60.4 ↑2.4 |
| + RL-CR | 79.1 ↑7.9 | 26.6 ↑2.8 | 38.9 ↑7.6 | 46.2 ↑1.4 | 30.2 ↑2.6 | 77.6 ↑41.8 | 61.1 ↑3.1 |
| **+ HiR (Ours)** | **83.6 ↑12.4** | **30.4 ↑6.6** | **41.8 ↑10.5** | **49.2 ↑4.4** | **31.7 ↑4.1** | **84.9 ↑49.1** | **63.6 ↑5.6** |
| Qwen2.5-7B-Instruct | 72.6 ↑0.0 | 26.2 ↑0.0 | 57.5 ↑0.0 | 49.4 ↑0.0 | 49.1 ↑0.0 | 51.4 ↑0.0 | 61.5 ↑0.0 |
| + SFT | 75.6 ↑3.0 | 27.9 ↑1.7 | 53.1 ↓4.4 | 48.2 ↓1.2 | 47.3 ↓1.8 | 67.8 ↑16.4 | 62.6 ↑1.1 |
| + DPO | 66.9 ↓5.7 | 25.9 ↓0.3 | 58.4 ↑0.9 | 50.6 ↑1.2 | 48.9 ↓0.2 | 56.5 ↑5.1 | **66.7** ↑5.2 |
| + RPO | 70.4 ↓2.2 | 26.5 ↑0.3 | 59.6 ↑2.1 | 51.0 ↑1.6 | 49.8 ↑0.7 | 61.3 ↑9.9 | 66.3 ↑4.8 |
| + RL-IR | 76.2 ↑3.6 | 31.1 ↑4.9 | 60.1 ↑2.6 | 49.8 ↑0.4 | 50.9 ↑1.8 | 72.2 ↑20.8 | 62.3 ↑0.8 |
| + RL-CR | 77.3 ↑4.7 | 31.6 ↑5.4 | 60.8 ↑3.3 | 51.2 ↑1.8 | 50.3 ↑1.2 | 73.5 ↑22.1 | 63.4 ↑1.9 |
| **+ HiR (Ours)** | **81.0 ↑8.4** | **35.8 ↑9.6** | **64.2 ↑6.7** | **54.6 ↑5.2** | **53.3 ↑4.2** | **79.4 ↑28.0** | 65.1 ↑3.6 |
| Qwen3-4B-Instruct-2507 | 83.4 ↑0.0 | 29.9 ↑0.0 | 67.5 ↑0.0 | 56.8 ↑0.0 | 57.7 ↑0.0 | 57.3 ↑0.0 | 76.1 ↑0.0 |
| + SFT | 83.4 ↑0.0 | 31.3 ↑1.4 | 64.2 ↓3.3 | 55.0 ↓1.8 | 55.9 ↓1.8 | 66.8 ↑9.5 | 74.9 ↓1.2 |
| + DPO | 83.9 ↑0.5 | 27.9 ↓2.0 | 68.0 ↑0.5 | 57.4 ↑0.6 | 58.1 ↑0.4 | 61.5 ↑4.2 | 78.0 ↑1.9 |
| + RPO | 84.7 ↑1.3 | 29.3 ↓0.6 | 69.1 ↑1.6 | 57.8 ↑1.0 | 58.9 ↑1.2 | 65.2 ↑7.9 | 78.2 ↑2.1 |
| + RL-IR | 85.0 ↑1.5 | 34.1 ↑4.2 | 69.8 ↑2.3 | 58.0 ↑1.2 | 58.2 ↑0.5 | 78.3 ↑21.0 | 77.6 ↑1.5 |
| + RL-CR | 85.8 ↑2.4 | 36.9 ↑7.0 | 68.5 ↑1.0 | 58.4 ↑1.6 | 59.6 ↑1.9 | 79.0 ↑21.7 | 78.2 ↑2.1 |
| **+ HiR (Ours)** | **86.3 ↑2.9** | **40.5 ↑10.6** | **73.2 ↑5.7** | **60.8 ↑4.0** | **61.5 ↑3.8** | **80.6 ↑23.3** | **80.4 ↑4.3** |

## 4.2. Main Results

**HiR applies to different model backbones and achieves consistent gains.** We conduct a comprehensive evaluation on seven instruction following benchmarks between our method and state-of-the-art baselines. As shown in Table 1, HiR achieves substantial improvements across different model backbones and scales, with Qwen3-4B-Instruct-2507 surpassing many leading LLMs (*e.g.*, Deepseek-V3.1, GPT-4.1) on multiple benchmarks. Under the RL framework, HiR delivers the best performance on most instruction following tasks, achieving greater gains than RL with constraint-level rewards (RL-CR). Moreover, our method exhibits superior robustness and generalization ability without observed performance degradation compared to SFT and DPO. Notably, HiR is particularly effective and yields larger improvements for initially weaker models, like Llama-3.2-3B-Instruct. We attribute this advantage to our hindsight replay mechanism that converts failure responses into successful ones, thus providing more informative learning signals. As the capability of the initial model increases, performance gains on saturated metrics (*e.g.*, Qwen3-4B-Instruct-2507 on IFEval) diminish, yet advantages remain pronounced on more challenging datasets, such as IFBench and MultiDimIF.

**HiR preserves general reasoning abilities in out-of-domain scenarios.** To assess whether optimizing for in-

*Table 2.* Performance of HiR on out-of-domain benchmarks.

| Model | MATH-500 | GPQA | MMLU-Pro |
|---|---|---|---|
| Llama-3.2-3B-Instruct | 47.8 | 30.8 | 34.9 |
| **+ HiR (Ours)** | 49.0 ↑1.2 | 29.5 ↓1.3 | 37.7 ↑2.8 |
| Qwen2.5-7B-Instruct | 76.6 | 36.4 | 56.3 |
| **+ HiR (Ours)** | 76.6 ↑0.0 | 35.9 ↓0.5 | 56.8 ↑0.5 |
| Qwen3-4B-Instruct-2507 | 86.8 | 61.8 | 69.6 |
| **+ HiR (Ours)** | 88.2 ↑1.4 | 62.1 ↑0.3 | 67.2 ↓2.4 |

struction following capability compromises broad problem-solving competence, we evaluate our method on three out-of-domain (OOD) reasoning benchmarks that are orthogonal to instruction following. As shown in Table 2, although HiR is trained solely on instruction following data, it preserves the models' OOD performances. Across all tested backbones, our method maintains parity with the initial models on these benchmarks, with no obvious drop and occasional marginal gains that fall within typical variance. These results reflect the robustness of our training data and indicate that HiR regularizes the policy toward better constraint satisfaction without collapsing general reasoning ability.

**HiR enhances both the sampling stability and reasoning boundaries.** Beyond Pass@1 scores, we analyze Pass@$k$ curves to characterize the reasoning boundary under increasing sampling budgets. As shown in Figure 3a, HiR consistently outperforms the initial model and RL-CR as $k$ grows,

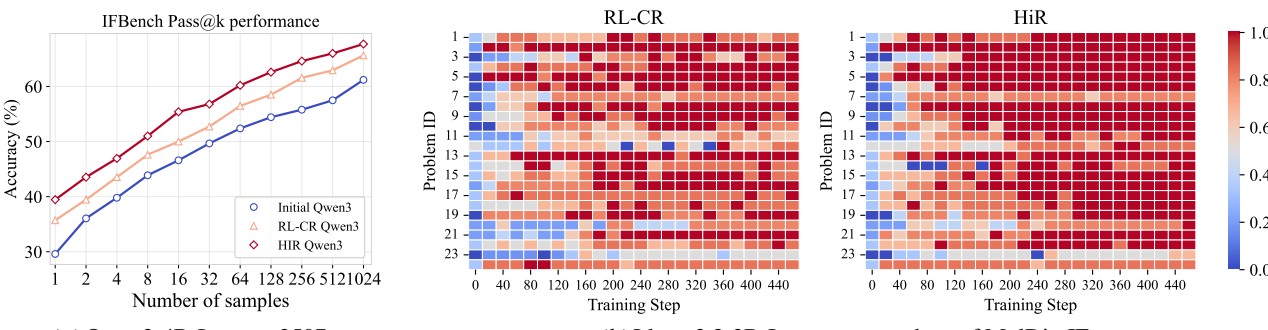

*(a)* Qwen3-4B-Instruct-2507.                                    *(b)* Llama3.2-3B-Instruct on a subset of MulDimIF.

*Figure 3.* (a) The pass@$k$ curves comparison after training, and (b) constraint-level accuracy heatmap comparison during training.

*Table 3.* Ablation study of selection strategy. **Bold** represents the best performance among all methods.

| Model | Method | IFEval | IFBench | CFBench | InfoBench | ComplexBench | MulDimIF | FollowBench |
|---|---|---|---|---|---|---|---|---|
| Llama-3.2-3B -Instruct | *w/* Random replay | 79.9 | 28.2 | 40.1 | 47.8 | 30.9 | 83.3 | 62.4 |
| | *w/* **HiR (Ours)** | **83.6** ↑3.7 | **30.4** ↑2.2 | **41.8** ↑1.7 | **49.2** ↑1.4 | **31.7** ↑0.8 | **84.9** ↑1.6 | **63.6** ↑1.2 |
| Qwen2.5-7B -Instruct | *w/* Random replay | 79.5 | 33.7 | 63.3 | 53.6 | 51.7 | 78.1 | **66.2** |
| | *w/* **HiR (Ours)** | **81.0** ↑1.5 | **35.8** ↑2.1 | **64.2** ↑0.9 | **54.6** ↑1.0 | **53.3** ↑1.6 | **79.4** ↑1.3 | 65.1 ↓1.1 |
| Qwen3-4B -Instruct-2507 | *w/* Random replay | 85.2 | 38.8 | 72.5 | 59.6 | 60.9 | 79.8 | 79.5 |
| | *w/* **HiR (Ours)** | **86.3** ↑1.1 | **40.5** ↑1.7 | **73.2** ↑0.7 | **60.8** ↑1.2 | **61.5** ↑0.6 | **80.6** ↑0.8 | **80.4** ↑0.9 |

demonstrating an expanded capability ceiling and improved sample efficiency. To better understand the learning dynamics and how its abilities evolve over time, we visualize the constraint-level accuracies on a subset of MultiDimIF across the training process for both HiR and RL-CR. The heatmap of HiR (Figure 3b) exhibits a smooth transition from low- to high-accuracy regions, indicating a consistent and stable improvement in instruction following capability rather than reliance on stochastic or sudden gains. Besides, we observe pronounced peaks for some problems, which suggests that HiR maintains the competence with minimal fluctuation once it masters a task. In contrast, the heatmap of RL-CR shows higher variability. While a few problems converge rapidly, others remain at fluctuating accuracy levels even after extensive training, revealing potential instability in its learning process. Overall, these analyses indicate that HiR delivers robust and consistent gains, leading to more reliable improvements while extending the achievable boundary.

*Table 4.* Performance under different optimization algorithms.

| Model | IFEval | IFBench | MulDimIF |
|---|---|---|---|
| Llama-3.2-3B-Instruct | 71.2 | 23.8 | 35.8 |
| **+ HiR (Reinforce++)** | 83.6 | 30.4 | 84.9 |
| **+ HiR (PPO)** | 84.1 | 31.3 | 86.5 |
| Qwen3-4B-Instruct-2507 | 83.4 | 29.9 | 57.3 |
| **+ HiR (Reinforce++)** | 86.3 | 40.5 | 80.6 |
| **+ HiR (PPO)** | 86.7 | 39.5 | 82.3 |

**HiR can be optimized using different RL algorithms.** In the main experiments, we adopt Reinforce++ as the policy optimization algorithm. Nevertheless, HiR is not tied to a specific RL algorithm and can be integrated with other policy optimization methods such as PPO (Schulman et al., 2017). In our PPO setting, the critic model is initialized

from the policy model and warmed up for 30 steps before joint actor-critic training. As shown in Table 4, HiR optimizing with PPO consistently improves instruction following performance under across different benchmarks, and outperforms Reinforce++ in most cases. These results demonstrate that the effectiveness of HiR does not depend on a particular optimization algorithm, but instead originates from its select-then-rewrite mechanism, which convert failed attempts into successful samples through to enrich the learning signals.

### 4.3. Ablation Study

**Selection Strategy.** We first analyze the contribution of selection strategies for HiR. Concretely, we adopt *Random Selection* strategy which arbitrarily picks a proportion of $k/m$ samples to replay. As shown in Table 3, our selection strategy performs optimally in most benchmarks. This confirms that not all failures are equally informative across different learning stages, and our efficiency can be attributed to a more adaptive selection of suitable samples for replay.

**Curriculum Schedule.** To understand how the trade-off between response diversity and constraint integrity impacts final performance, we plot benchmark accuracy training with different initial curriculum weight $\lambda_0$. As shown in Figure 4, HiR outperforms the baseline RL-CR (in Table 1) over a wide range, with pronounced performance degradation only when $\lambda_0$ is excessively small or large. This phenomenon highlights a trade-off between exploration and exploitation: emphasizing constraint integrity too early (large $\lambda_0$) may lead to insufficient exploration of the solution space; while prioritizing response diversity overlong (small $\lambda_0$) may fail to provide the necessary guidance required to satisfy all constraints in the later training stage. Notably, we observe that

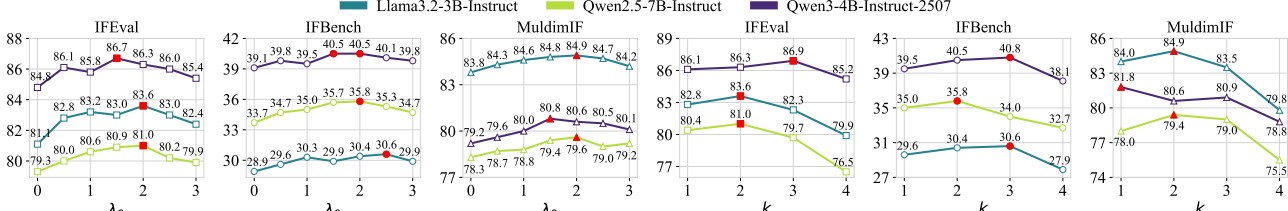

*Figure 4.* Ablation study of the initial curriculum weight and replay numbers, with red markers indicating the best performance.

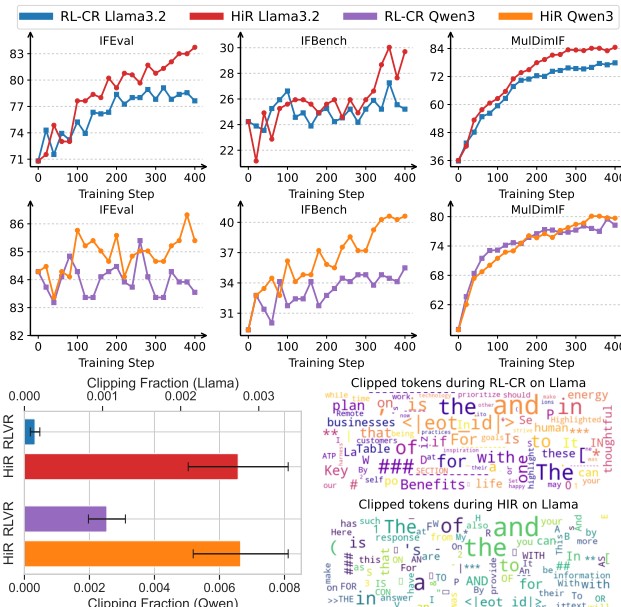

*Figure 5.* Training dynamics of different model backbones. HiR exhibits higher training efficiency than baseline RL-CR.

the optimal performance is stably located around $\lambda_0 = 2$ across various model backbones and tasks, which demonstrates that our method is robust rather than overly sensitive to hyperparameter choices.

**Replayed Numbers.** We further investigate the ideal replayed number $k$, with $m + k$ controlled to 8. In most cases, $k = 2$ or 3 yields the best results. Increasing $k$ beyond 3 leads to performance degradation, as the proportion of on-policy data in the training batch becomes relatively low.

## 5. In-Depth Analysis

**Training Dynamics.** We report the performance on different benchmarks and clipping ratio during training. Figure 5 demonstrates that the training process with HiR remains stable and shows superior training efficiency compared to vanilla RL, achieving better benchmark performance under the same consumed prompts and **fewer sampling budgets**. A more interesting observation is that despite clipping more tokens due to replay and therefore using fewer for training, HiR still achieves higher training efficiency. This finding further reveals that token-level gradient estimates may be inherently noisy and inefficient for sample exploitation. For

example, the clipped tokens during RL-CR contain some key information relevant to the instructions like substantive words "plan" and "benefits". In contrast, HiR tends to clip the gradient of less informative transitional or connective tokens, providing a more reliable and effective signal.

**Parameter Change.** To investigate the underlying sources of performance gain, we conduct a parameter-level analysis following Ye et al. (2025). We quantified the relative change rate $|W_{\text{Init}} - W_{\text{HiR}}|/|W_{\text{Init}}|$ in model parameters after HiR training, and group the values by different modules. As depicted in Figure 6, we observe that most significant updates occurred within the value modules of self-attention. This suggests that HiR primarily optimizes how the model "attends" to given information. Moreover, these variations were uniformly distributed across all layers, indicating a global rather than local adjustment. Therefore, the improvement of HiR may stem from an enhanced capacity to identify and exploit critical input tokens during training, thereby boosting its instruction following performance.

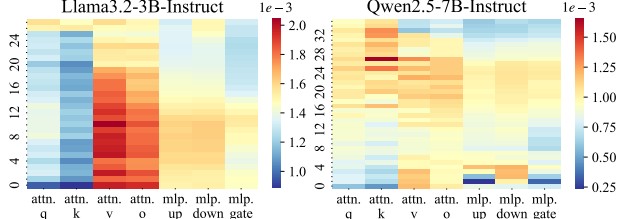

*Figure 6.* Parameter change over each module after HiR training.

**Attention Attribution.** To provide deeper insights into the evolution of attention mechanisms, we compute the average attention allocated to each input token during generation. As shown in Table 5, HiR training drives a more pronounced increase in attention toward constraint-related tokens than RL-CR, while simultaneously diminishing attention toward irrelevant tokens. This suggests that the model has a refined discriminative capability to identify critical constraint information while suppressing noise from distracting elements. These qualitative results empirically validate the superiority of our HiR framework, guiding the model toward more robust performance gains compared to vanilla RL approaches.

## 6. Related Work

**Instruction Following Methods.** Recent advancements in complex instruction following have shifted from static data

*Table 5.* A visualization of average attention allocated to each input token during generation, with `darker` representing greater attention.

| Case (Initial) | Case (*w/* RL-CR) | Case (*w/* HiR) |
|---|---|---|
| *Llama-3.2-3B-Instruct: After tuning by HiR, the model imposes greater attention to constraint 'capital letters' and content of the given sentence, ensuring both format adherence and content coherence.* | | |

```
Write a paragraph critique of the following sentence
in all  capital letters, no lowercase letters allowed:
"If  the law is bad, you should not follow it". Label
each paragraph with PAR AGR APH X.
```
```
Write a 2 paragraph critique of the following sentence
in all  capital letters, no lowercase letters allowed:
"If  the law is bad, you should not follow it". Label
each paragraph with PAR AGR APH X.
```
```
Write a 2 paragraph critique of the following sentence
in all  capital letters, no lowercase letters allowed:
"If  the law is bad, you should not follow it". Label
each paragraph with PAR AGR APH X.
```

*Qwen3-4B-Instruct-2507: After tuning by HiR, the model places more emphasis on keywords 'compensated' and 'immigrants' while reducing the attention to less informative pronoun 'me', enabling to concentrate on the key information.*

```
Could you please give me the pros and cons of working
abroad wrapped  only  in  JSON format. Please  also
make  sure  to  include keywords 'compensated'  and
'immigrants'  in  the response.
```
```
Could you please give me the pros and cons of working
abroad wrapped  only  in  JSON format. Please  also
make  sure  to  include keywords 'compensated'  and
'immigrants'  in  the response.
```
```
Could you please give me the pros and cons of working
abroad  wrapped  only  in  JSON format. Please  also
make  sure  to  include keywords 'compensated'  and
'immigrants'  in  the response.
```

synthesis methods like iterative refinement (Xu et al., 2024; Liu et al., 2025b; Cheng et al., 2025), back-translation (Qi et al., 2025b) and preference optimization (Liu et al., 2025a) for instruction tuning, to reinforcement learning with verifiable rewards (RLVR) (Lambert et al., 2024; Peng et al., 2025; Qin et al., 2025; Dong et al., 2025), which utilizes rule or code-based verifiers to provide reward signals. To address the nuances of multi-constraint alignment, methods like IOPO (Zhang et al., 2025e) and RPO (Huang et al., 2025b) construct fine-grained preference pairs via adversarial inputs or constraint reversal. However, these approaches face efficiency bottlenecks in the RL paradigm. Specifically, IOPO mandates perfect positive samples, hindering sampling efficiency for weaker models, while RPO relies on the strong assumption that all constraints are logically reversible, limiting its applicability to implicit constraints. In contrast, HiR bypasses these limitations by simply removing unmet constraints to retrospectively map failed trajectories to satisfied pseudo-instructions, solving the sparse reward problem with superior sample efficiency.

**Hindsight Experience Replay.** Hindsight Experience Replay (HER) is a technique in traditional reinforcement learning (Liu et al., 2024b; 2023) designed to mitigate sparse rewards and reduce the need for complex reward engineering. Andrychowicz et al. (2017) first introduces HER to replay failed experiences by replacing original goals with achieved states in the environment. On top of HER, several subsequent works have been proposed to encourage better exploration in environment (Fang et al., 2019; Liu et al., 2019), and identify trajectories with higher energy to benefit training (Zhao & Tresp, 2018; Nguyen et al., 2019). DHER (Fang et al., 2018) further extends training from static goals to complex dynamic goal settings. More recently, several works (Li et al., 2025c; Zhang et al., 2025a; Zhan et al., 2026) have demonstrated the effectiveness of experience replay mechanisms in reinforcement learning for LLMs. However, prior HER-based methods have not been explored in RL training of LLMs yet as the states in LLMs are high-dimensional and semantically coherent token se-

quences, lacking quantifiable representations for naive goal replacement. In this work, HiR treats atomic constraints as hindsight goals in instruction space, coupled with an adaptive replay mechanism that trades off response diversity and constraint integrity alongside the model's learning progress.

## 7. Conclusion

This work proposes HiR, a simple and efficient method to incentivize the capability of LLMs for solving complex instructions. HiR employs a *select*-then-*rewrite* strategy that adaptively selects failure samples in a curriculum-based manner, followed by rewriting their instructions into "hindsight" pseudo-instructions for replay. In this way, HiR implicitly introduces an instruction-wise preference into the RL training objective, enabling LLMs to precisely identify unmet constraints in instructions for effective learning with only a binary reward. Extensive experiments demonstrate that HiR consistently outperforms current baselines and achieves competitive results compared with leading models. Currently we apply hindsight instruction replay to RL for LLMs, we expect to explore applications to multi-modal tasks and agentic scenarios for future work.

## Acknowledgements

This work is supported in part by the Hangzhou Joint Funds of the Zhejiang Provincial Natural Science Foundation of China under Grant No. LHZSD24F020001, in part by the Fundamental Research Funds for the Central Universities under Grant No. 226-2025-00057, and in part by the advanced computing resources provided by the Supercomputing Center of Hangzhou City University.

## Impact Statement

This paper presents work whose goal is to advance the field of instruction following capability in LLMs. There are many potential societal consequences of our work, none of which we feel must be specifically highlighted here.

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

# Appendix

## A. More Experimental Results and Discussion

### A.1. Implementation Details

All experiments run on 8×A100-80GB GPUs. We use LLaMA-Factory (Zheng et al., 2024) for SFT and DPO training, and verl (Sheng et al., 2025) for RL training. The detailed training configurations of SFT and DPO are provided in Table 6a, and the training configurations of RL are provided in Table 6b. Note that for a fair comparison, we ensure the consistent number of training samples in a batch across RL methods, with less sampling budget of HiR (generate 6 responses then replay 2 responses) compared to RL-IR and RL-CR (generate 8 responses).

*Table 6.* Training configurations across different methods and model backbones.

*(a)* Training configuration of SFT and DPO.

| Method | SFT, DPO, RPO |
|---|---|
| **Training** | per_device_train_batch_size = 16, gradient_accumulation_steps = 16 |
| | learning_rate = 1e-6, lr_scheduler_type = constant |
| | cutoff_len = 4096, warmup_steps = 10, epochs = 5 |
| **Optimizations** | deepspeed: z3, bf16 |

*(b)* Training configuration of RL.

| Settings | Hyperparameters |
|---|---|
| **HiR** | $m = 6, k = 2, \eta = 0.05, \lambda_0 = 2$ |
| **RL-IR, RL-CR** | rollout_n = 8 ($m = 8, k = 0$) |
| **Sampling** | top_k = -1, top_p = 1.0, temperature = 1.0 |
| | max_prompt_length = 2,048, max_response_length = 4,096 |
| **Training** | train_batch_size = 64, ppo_mini_batch_size = 32 |
| | ppo_micro_batch_size_per_gpu = 8, log_prob_micro_batch_size_per_gpu = 8 |
| | learning_rate = 1e-6, kl_loss_coef = 1e-4, epochs = 5 |
| **Optimizations** | param_offload, flash_attn, bf16 |

We use the vLLM (Kwon et al., 2023) engine to generate responses and the API of Deepseek-V3.1 for evaluation. The generation temperature is set to 0.6, and the maximum output length is set to 4,096 tokens. We report the average of five independent evaluation results across all benchmarks. For instruction following tasks, we use the default prompt template of models in evaluation. For OOD tasks (*i.e.*, MATH-500, GPQA, MMLU-Pro), we add additional CoT prompts in evaluation as shown in Table 7.

*Table 7.* Evaluation prompts on initial model across out-of-domain benchmarks.

| Datasets | CoT Prompts |
|---|---|
| MATH-500 | Question: {}\nPlease reason step by step, and put your final answer within \boxed{}. |
| GPQA & MMLU-Pro | Question: {}\nAnswer the multiple choice question. The last line of your response should be of the following format: 'Answer: $LETTER' (without quotes) where LETTER is one of choices. Think step by step before answering. |

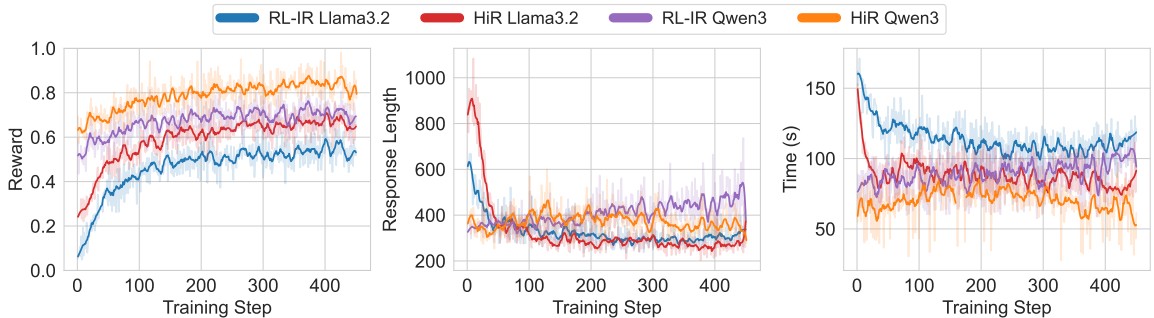

*Figure 7.* Training dynamics of reward, response length and time across different methods.

## A.2. Additional Results

### A.2.1. TRAINING DYNAMICS

We further present the dynamics of time, reward, and response length during training. Figure 7 demonstrates that our method is more efficient than vanilla RL methods in terms of time benefiting from the replay strategy. This is because HiR requires fewer rollouts, which reduces both the sampling time and reward calculation time. The additional rewriting process is a rule-based string modification (removing unmet constraints based on verifier feedback), which is computationally negligible compared to the gradient computation or the forward pass of the LLMs.

### A.2.2. IMPORTANCE SAMPLING ANALYSIS

When training a policy $\pi_\theta$ using original samples collected from the old policy $\pi_{old}$, importance sampling provides an unbiased estimator for the policy gradient. The key difference in our setting is that we conduct selection for the replayed samples, which changes the original probability of a response that will be selected. Therefore, the importance sampling ratio $\rho_{t,\theta}^{\prime(i)} = \frac{\pi_\theta(y_t^{\prime(i)}|q^{\prime(i)},y_{<t}^{\prime(i)})}{\pi_{old}(y_t^{\prime(i)}|q,y_{<t}^{\prime(i)})}$ in Eq. (9) should not be interpreted as an unbiased correction in traditional RL but acts as a PPO-inspired weighting mechanism. Intuitively, the selection plays a role similar to curriculum learning, prioritizing more informative samples and improving learning efficiency. More broadly, such selection for experience replay is common in LLM post-training (Li et al., 2025c; Zhang et al., 2025a; Zhan et al., 2026). Although strict importance correction is typically not enforced, its empirical effectiveness is well established. In this context, HiR follows a similar paradigm.

To further investigate this design, we modify our original training objective to remove selection bias. Specifically, we select $k = 2$ samples for replay according to the score-based probability in a sampling group $\mathcal{G}$. The probability of response $y_i$ being selected is $P(y_i) = \frac{F(y_i)}{\sum_{y \in \mathcal{G}} F(y)}$, where $F(y)$ denotes the score of response $y$ calculated by response diversity and constraint integrity. We additionally introduce a weights $w^{\prime(i)} = \frac{1}{m} \cdot \frac{1}{P(y_i)}$ in the objective to correct the selection bias, which preserves the expected gradient mathematically:

$$\mathcal{J}_{\text{HiR-Weighted}}(\theta) = \mathbb{E}_{q \sim \mathcal{D}, \{y^{(i)}\}_{i=1}^m \sim \pi_{old}(\cdot|q), \{q^{\prime(i)}, y^{\prime(i)}\}_{i=1}^k \sim \mathcal{H}}$$

$$\underbrace{\left[ \frac{1}{m} \sum_{i=1}^m \frac{1}{|y^{(i)}|} \sum_{t=1}^{|y^{(i)}|} \min\left(\rho_{t,\theta}^{(i)} A_t^{(i)}, \text{clip}(\rho_{t,\theta}^{(i)}, 1 \pm \epsilon) A_t^{(i)}\right)}_{\text{Objective for Initial Samples}} + \underbrace{\frac{1}{k} \sum_{i=1}^k \frac{w^{\prime(i)}}{|y^{\prime(i)}|} \sum_{t=1}^{|y^{\prime(i)}|} \min\left(\rho_{t,\theta}^{\prime(i)} A_t^{\prime(i)}, \text{clip}(\rho_{t,\theta}^{\prime(i)}, 1 \pm \epsilon) A_t^{\prime(i)}\right)}_{\text{Objective for Replayed Samples}} \right].$$

$$(12)$$

*Table 8.* Comparison of different importance ratio correction methods. **Bold** represents the best results.

| Objective | IFEval | IFBench | CFBench | InfoBench | ComplexBench | MulDimIF | FollowBench |
|---|---|---|---|---|---|---|---|
| *Model: Llama-3.2-3B-Instruct* | | | | | | | |
| $\mathcal{J}_{\text{HiR-Weighted}}$ | 82.3 | 29.6 | **42.1** | 48.4 | **32.2** | 84.2 | 62.5 |
| $\mathcal{J}_{\text{HiR}}$ | **83.6** | **30.4** | 41.8 | **49.2** | 31.7 | **84.9** | **63.6** |
| *Model: Qwen3-4B-Instruct-2507* | | | | | | | |
| $\mathcal{J}_{\text{HiR-Weighted}}$ | **86.3** | 40.1 | 73.0 | **61.4** | **61.9** | 80.1 | **80.8** |
| $\mathcal{J}_{\text{HiR}}$ | **86.3** | **40.5** | 73.2 | 60.8 | 61.5 | **80.6** | 80.4 |

The experimental results in Table 8 reveal insights about the trade-offs between theoretical correctness and practical performance. Although $\mathcal{J}_{\text{HiR-Weighted}}$ is theoretically unbiased, it does not demonstrate superior results compared to the current objective. Moreover, correcting selection bias introduces higher variance in gradient, which may offset its theoretical advantages. This indicates the objective $\mathcal{J}_{\text{HiR}}$ may effectively act as a proximal policy update that encourages the model to learn from the selected informative failures.

### A.2.3. MORE BASELINES

We further compare the performance of HiR with self-refinement method HeRL (Zhang et al., 2026) and dense reward training algorithms TreeRPO (Yang et al., 2025b). HeRL uses the unmet rubrics in failed attempts to guide revisions for RL training, and TreeRPO provides fine-grained and dense reward signals based on step-level groups generated during tree sampling. As shown in Table 9, HiR still yields better performance under less computational budget.

*Table 9.* Comparison with HeRL and TreeRPO using model Qwen3-4B-Instruct-2507.

| Method | IFEval | IFBench | MulDimIF |
|---|---|---|---|
| Initial | 83.4 | 29.9 | 57.3 |
| + HeRL | 86.1 | 38.8 | 80.2 |
| + TreeRPO | 86.1 | 37.8 | 79.8 |
| **+ HiR (Ours)** | **86.3** | **40.5** | **80.6** |

### A.3. Limitations and Future Work

Currently, we primarily focus on instruction following tasks. While HiR demonstrates promising results compared to vanilla RL baselines, several limitations remain that highlight directions for future research. Firstly, although we employ a hybrid method that combines rule-based and LLM-based scoring for rewards, precise evaluation remains challenging. Using smaller models for scoring may introduce biases and lead to suboptimal outcomes, while using larger models results in greater computational costs. Therefore, it is necessary to develop an open source, moderately-sized reward model that provides unbiased judgment. Secondly, our method relies on high-quality instructions with decomposable atomic constraints to facilitate training, while such datasets are still scarce in the community. As a preliminary exploration of applying hindsight replay for LLMs, we simplify the potential relationships among constraints and treat them independently for convenience. This premise is consistent with popular instruction-following benchmarks like IFEval and IFBench, where constraints are explicitly designed as independent, verifiable atomic items. Therefore, possible logical contradictions may be introduced by naively removing unmet constraints when they have complex relationships. In the future, we can extend HiR training data with structured constraint dependency to handle complex nested, dependent, and prioritized constraints. Furthermore, we expect to systematically develop pipelines for data production and explore applications to agentic scenarios in RL.

### A.4. Discussions

**Rewriting method.** In this work, we adopt a deterministic and lightweight hindsight rewriting solution just by removing the unmet constraints, which avoids additional rewriting noise or hallucinated constraints introduced by LLMs. This provides a simple yet stable way to validate the effectiveness of our proposed HiR. Beyond constraint removal, we can refine the unmet constraint to match the response using LLMs, rather than removing it entirely. For example, if the original instruction contains "write a poem about spring" and the response is a poem about summer, we can rewrite the instruction to "write a poem about summer" using LLMs, which retains more constraint information while maintaining the validity of the positive sample after rewriting. Thus, more flexible rewriting methods, such as re-expressing constraints, are promising directions.

**Diversity metrics $F_{div}$.** In this work, we employ entropy that measures distribution-level diversity as $F_{div}$. This design is motivated by several works (Wang et al., 2026; Cui et al., 2025) that reveal high-entropy tokens (distribution uncertainty) play an important role in RL training, which is more informative in reasoning. Nevertheless, entropy represents one possible formulation of diversity in instruction following. Alternative definitions may further improve the effectiveness of diversity-aware optimization. For instance, diversity could be characterized through heterogeneous constraint-violation patterns across responses in a sampling group, encouraging the model to explore different failure modes during training.

## B. Proof and Analysis

### B.1. Technical Settings and Notations

**Settings**. Since the order of samples does not affect subsequent analysis, we assume that samples with indices from $i = 1$ to $i = k$ are the failed samples used to replay for convenience. Our theoretical settings involve two main simplifications on Eq. (8). First, we omit the clipping operation, because the clipping mechanism primarily serves as a practical stabilization

heuristic to limit excessively large policy updates. Tokens that are out of range will not contribute to gradient, so the omission of the clipping does not affect the trajectory-level analysis. Second, we omit the nuanced differences in advantages among tokens within a response, as the KL coefficient is relatively small and will not be dominant.

**Notations**. We use $\pi_\theta$ to denote Large Language Models (LLMs) parameterized by $\theta$. The response $y^w$ and $y^l$ denote the winning (positive) and losing (negative) responses, and $y^r$ denotes the responses that are selected for replay.

## B.2. Proof of Proposition 3.2

*Proof.* The HiR training objective that omits the clipping mechanism is:

$$
\begin{aligned}
\mathcal{J}_{\text{HiR}}(\theta) =& \mathbb{E}_{\substack{q \sim \mathcal{D} \\ \{y^{(i)}\}_{i=1}^m \sim \pi_{\text{old}}(\cdot|q) \\ \{q'^{(i)}, y'^{(i)}\}_{i=1}^k \sim \mathcal{H}}} \left[ \frac{1}{m} \sum_{i=1}^m \frac{1}{|y^{(i)}|} \sum_{t=1}^{|y^{(i)}|} \rho_{t,\theta}^{(i)} A_t^{(i)} + \frac{1}{k} \sum_{i=1}^k \frac{1}{|y'^{(i)}|} \sum_{t=1}^{|y'^{(i)}|} \rho_{t,\theta}'^{(i)} A_t'^{(i)} \right] \\
=& \mathbb{E}_{\substack{q \sim \mathcal{D} \\ \{y^{(i)}\}_{i=1}^m \sim \pi_{\text{old}}(\cdot|q) \\ \{q'^{(i)}, y'^{(i)}\}_{i=1}^k \sim \mathcal{H}}} \left[ \frac{1}{m} \sum_{i=1}^m \frac{1}{|y^{(i)}|} \sum_{t=1}^{|y^{(i)}|} \frac{\pi_\theta(y_t^{(i)} \mid q, y_{<t}^{(i)})}{\pi_{\text{old}}(y_t^{(i)} \mid q, y_{<t}^{(i)})} A_t^{(i)} + \frac{1}{k} \sum_{i=1}^k \frac{1}{|y'^{(i)}|} \sum_{t=1}^{|y'^{(i)}|} \frac{\pi_\theta(y_t'^{(i)} \mid q'^{(i)}, y_{<t}'^{(i)})}{\pi_{\text{old}}(y_t^{(i)} \mid q, y_{<t}^{(i)})} A_t'^{(i)} \right].
\end{aligned}
\tag{13}
$$

According to the standard importance sampling formula $\mathbb{E}_q \left[ \frac{p(x)}{q(x)} f(x) \right] = \mathbb{E}_p[f(x)]$ and the log-derivative, we can obtain the gradient as:

$$
\begin{aligned}
\nabla \mathcal{J}_{\text{HiR}}(\theta) =& \mathbb{E}_{q \sim \mathcal{D}, \{y^{(i)}\}_{i=1}^m \sim \pi_\theta(\cdot|q), \{q'^{(i)}, y'^{(i)}\}_{i=1}^k \sim \mathcal{H}} \\
& \left[ \frac{1}{m} \sum_{i=1}^m \frac{1}{|y^{(i)}|} \sum_{t=1}^{|y^{(i)}|} \nabla \log \pi_\theta(y_t^{(i)} \mid q, y_{<t}^{(i)}) A_t^{(i)} + \frac{1}{k} \sum_{i=1}^k \frac{1}{|y'^{(i)}|} \sum_{t=1}^{|y'^{(i)}|} \nabla \log \pi_\theta(y_t'^{(i)} \mid q'^{(i)}, y_{<t}'^{(i)}) A_t'^{(i)} \right].
\end{aligned}
\tag{14}
$$

By separating the failed responses used for replay (*i.e.*, indices from 1 to $k$) from original samples and based on the fact that $\{y'^{(i)}\}_{i=1}^k = \{y^{(i)}\}_{i=1}^k$, we can derive:

$$
\begin{aligned}
\nabla \mathcal{J}_{\text{HiR}}(\theta) =& \mathbb{E}_{q \sim \mathcal{D}, \{y^{(i)}\}_{i=1}^m \sim \pi_\theta(\cdot|q), \{q'^{(i)}\}_{i=1}^k \sim \mathcal{H}} \\
& \left[ \frac{1}{m} \sum_{i=k}^m \frac{1}{|y^{(i)}|} \sum_{t=1}^{|y^{(i)}|} \nabla \log \pi_\theta(y_t^{(i)} \mid q, y_{<t}^{(i)}) A_t^{(i)} + \right. \\
& \left. \frac{1}{m} \sum_{i=1}^k \frac{1}{|y^{(i)}|} \sum_{t=1}^{|y^{(i)}|} \nabla \log \pi_\theta(y_t^{(i)} \mid q, y_{<t}^{(i)}) A_t^{(i)} + \frac{1}{k} \sum_{i=1}^k \frac{1}{|y^{(i)}|} \sum_{t=1}^{|y^{(i)}|} \nabla \log \pi_\theta(y_t^{(i)} \mid q'^{(i)}, y_{<t}^{(i)}) A_t'^{(i)} \right].
\end{aligned}
\tag{15}
$$

We further divide the first term into two groups: positive (winning) and negative (losing) samples, which obtains:

$$
\begin{aligned}
\nabla \mathcal{J}_{\text{HiR}}(\theta) =& \mathbb{E}_{q \sim \mathcal{D}, \{y^{(i)}\}_{i=1}^m \sim \pi_\theta(\cdot|q), \{q'^{(i)}\}_{i=1}^k \sim \mathcal{H}} \\
& \left[ \frac{1}{m} \left( \sum_{i=k}^{G^-} \frac{1}{|y^{(i)}|} \sum_{t=1}^{|y^{(i)}|} A^- \nabla \log \pi_\theta(y_t^{(i)} \mid q, y_{<t}^{(i)}) + \sum_{i=G^-}^m \frac{1}{|y^{(i)}|} \sum_{t=1}^{|y^{(i)}|} A^+ \nabla \log \pi_\theta(y_t^{(i)} \mid q, y_{<t}^{(i)}) \right) + \right. \\
& \left. \left( \frac{1}{m} \sum_{i=1}^k \frac{1}{|y^{(i)}|} \sum_{t=1}^{|y^{(i)}|} A^- \nabla \log \pi_\theta(y_t^{(i)} \mid q, y_{<t}^{(i)}) + \frac{1}{k} \sum_{i=1}^k \frac{1}{|y^{(i)}|} \sum_{t=1}^{|y^{(i)}|} A'^+ \nabla \log \pi_\theta(y_t^{(i)} \mid q'^{(i)}, y_{<t}^{(i)}) \right) \right] \\
=& \mathbb{E}_{q \sim \mathcal{D}, \{y^{(i)}\}_{i=1}^m \sim \pi_\theta(\cdot|q), \{q'^{(i)}\}_{i=1}^k \sim \mathcal{H}} \\
& \left[ \left( \alpha_1 \cdot \frac{1}{m - G^-} \sum_{i=G^-}^m \frac{1}{|y^{(i)}|} \sum_{t=1}^{|y^{(i)}|} \nabla \log \pi_\theta(y_t^{(i)} \mid q, y_{<t}^{(i)}) - \beta_1 \cdot \frac{1}{G^- - k} \sum_{i=k}^{G^-} \frac{1}{|y^{(i)}|} \sum_{t=1}^{|y^{(i)}|} \nabla \log \pi_\theta(y_t^{(i)} \mid q, y_{<t}^{(i)}) \right) + \right. \\
& \left. \left( \alpha_2 \cdot \frac{1}{k} \sum_{i=1}^k \frac{1}{|y^{(i)}|} \sum_{t=1}^{|y^{(i)}|} \nabla \log \pi_\theta(y_t^{(i)} \mid q'^{(i)}, y_{<t}^{(i)}) - \beta_2 \cdot \frac{1}{k} \sum_{i=1}^k \frac{1}{|y^{(i)}|} \sum_{t=1}^{|y^{(i)}|} \nabla \log \pi_\theta(y_t^{(i)} \mid q, y_{<t}^{(i)}) \right) \right],
\end{aligned}
\tag{16}
$$

where $\alpha_1 = \frac{m-G^-}{m}A^+, \beta_1 = -\frac{G^--k}{m}A^-, \alpha_2 = A'^+, \beta_2 = -\frac{k}{m}A^-$. Note that they are all positive values as $A^+ > 0$ and $A^- < 0$.

By the law of large numbers $\lim_{N\to\infty} \frac{1}{N}\sum_{i=1}^N f(y) = \mathbb{E}_{y\in\pi_\theta} f(y)$, the empirical mean of finite samples converges to the true expectation as the sample size $N \to \infty$. We use $y^r$ to denote the negative samples selected for replay, $y^w$ to denote the positive samples, and $y^l$ to denote the negative samples that are not used for replay. We thus reformulate the empirical objective as the expected training objective:

$$\nabla\mathcal{J}_{\text{HiR}}(\theta) = \mathbb{E}_{q\sim\mathcal{D},q'\sim\mathcal{H}}$$

$$\left[\left(\alpha_1 \cdot \mathop{\mathbb{E}}_{y^w\sim\pi_\theta(\cdot|q)}\frac{1}{|y^w|}\sum_{t=1}^{|y^w|}\nabla\log\pi_\theta(y_t^w \mid q, y_{<t}^w) - \beta_1 \cdot \mathop{\mathbb{E}}_{y^l\sim\pi_\theta(\cdot|q)}\frac{1}{|y^l|}\sum_{t=1}^{|y^l|}\nabla\log\pi_\theta(y_t^l \mid q, y_{<t}^l)\right) + \right.$$

$$\left.\left(\alpha_2 \cdot \mathop{\mathbb{E}}_{y^r\sim\pi_\theta(\cdot|q)}\frac{1}{|y^r|}\sum_{t=1}^{|y^r|}\nabla\log\pi_\theta(y_t^r \mid q', y_{<t}^r) - \beta_2 \cdot \mathop{\mathbb{E}}_{y^r\sim\pi_\theta(\cdot|q)}\frac{1}{|y^r|}\sum_{t=1}^{|y^r|}\nabla\log\pi_\theta(y_t^r \mid q, y_{<t}^r)\right)\right]. \quad (17)$$

Therefore, the final expected training objective of HiR can be written as a form of preference learning on both the response- and instruction-level:

$$\nabla J_{\text{HiR}}(\theta) = \mathbb{E}_{q\sim\mathcal{D},q'\sim\mathcal{H}}$$

$$\left[\underbrace{\left(\alpha_1\mathop{\mathbb{E}}_{y^w\sim\pi_\theta(\cdot|q)}\nabla\log\pi_\theta(y^w|q) - \beta_1\mathop{\mathbb{E}}_{y^l\sim\pi_\theta(\cdot|q)}\nabla\log\pi_\theta(y^l|q)\right)}_{\text{Response-level Preference}} + \underbrace{\left(\alpha_2\mathop{\mathbb{E}}_{y^r\sim\pi_\theta(\cdot|q)}\nabla\log\pi_\theta(y^r|q') - \beta_2\mathop{\mathbb{E}}_{y^r\sim\pi_\theta(\cdot|q)}\nabla\log\pi_\theta(y^r|q)\right)}_{\text{Instruction-level Preference}}\right]$$

$$(18)$$

where we define the length normalized $\nabla\log\pi_\theta(y \mid q) = \frac{1}{|y|}\sum_{t=1}^{|y|}\nabla\log\pi_\theta(y_t \mid y_{<t}, q)$. $\square$

## C. Dataset Information

### C.1. Training Dataset

To facilitate hindsight rewriting, we construct 16,969 queries with decomposable constraints in different scenarios. Specifically, we collect public data from various sources, including training set of MulDimIF (Ye et al., 2025), VerIF (Peng et al., 2025), IFTrain (Pyatkin et al., 2025), and Chatbot Arena (Zheng et al., 2023). We first break down the atomic constraints in the instruction to form a constraint set $\mathcal{C}$ and then filter instructions with less than 5 atomic constraints. We further synthesize constraints using programmatic approaches to enrich the dataset. Finally, we obtained the `HIR-16K` dataset with 76,456 hard constraints and 46,536 soft constraints (a ratio of 1.6:1). Figure 8 shows the detailed composition and distribution of constraints in the training dataset.

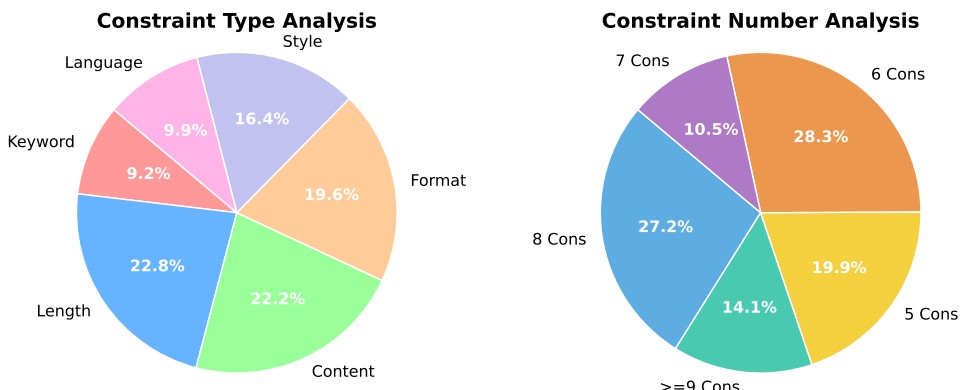

*Figure 8.* (Left) Proportion of constraint types in our training dataset. (Right) Proportion of constraint numbers per instruction.

## C.2. Evaluation Dataset

**IFEval** (Zhou et al., 2023b) is a benchmark for evaluating the instruction following ability of LLMs, focusing on a set of verifiable instructions. The dataset comprises 25 types of verifiable instructions and 541 prompts, with each prompt including one or more verifiable instructions, such as word-count constraints and keyword occurrence requirements.

**IFBench** (Pyatkin et al., 2025) is designed to evaluate the precise instruction following generalization of LLMs, aiming to test whether models can generalize to previously unseen instruction types. The dataset contains 58 new verifiable constraints with corresponding verification functions and 294 prompts, covering word-count limits, formatting requirements, counting, copying, and sentence/word/character manipulations. Each prompt may include one or more constraints.

**CFBench** (Zhang et al., 2025d) is a comprehensive Chinese benchmark comprising 1,000 carefully curated samples, covering over 200 real-world scenarios and more than 50 natural language processing tasks. Each sample contains multiple constraints organized into 10 primary categories and over 25 subcategories, with constraints seamlessly integrated into the original instructions and complex combinations carefully handled. The benchmark uses a multi-dimensional evaluation framework with requirement prioritization to assess performance from multiple perspectives.

**InfoBench** (Qin et al., 2024) comprises 500 diverse instructions and 2,250 decomposed questions across multiple constraint categories, designed to test and analyze the instruction following capabilities of LLMs systematically. The constraints involved in each instruction are categorized into five types: Content, Linguistic, Style, Format, and Number.

**ComplexBench** (Wen et al., 2024) is designed to evaluate the ability of LLMs to follow complex instructions under different compositions of constraints. The dataset is built upon a hierarchical taxonomy of 1,150 complex instructions, encompassing 4 constraint types, 19 constraint dimensions, and 4 composition types.

**MulDimIF** (Ye et al., 2025) is an instruction following benchmark built upon a multi-dimensional constraint framework. It covers three constraint patterns, four constraint categories, and four difficulty levels, comprising 1,200 code-verifiable instruction following test samples. MulDimIF enables systematic and fine-grained evaluation of large language models under diverse constraint forms and varying levels of complexity.

**FollowBench** (Jiang et al., 2024) is a multi-level, fine-grained instruction following benchmark for LLMs, designed to systematically evaluate their ability to understand and execute constraints in real-world instruction scenarios. The benchmark explicitly models constraint elements from user instructions, covering 5 types of fine-grained constraints: Content, Situation, Style, Format, and Example.

**MATH-500** (Lightman et al., 2024) dataset contains 500 high school–level math problems, covering 7 major areas such as precalculus, algebra, number theory, counting & probability, geometry, intermediate algebra, and precalculus.

**GPQA** (Rein et al., 2024) is a challenging scientific multiple-choice question dataset, primarily authored by experts in biology, physics, and chemistry, comprising 448 questions in the main set. The questions are carefully curated to ensure both high expertise and difficulty.

**MMLU-Pro** (Wang et al., 2024) is a benchmark for advanced multi-disciplinary language understanding and reasoning, designed to comprehensively evaluate LLMs on complex, multi-domain tasks. The dataset spans 14 disciplines, including mathematics, physics, chemistry, law, engineering, psychology, and health, comprising 12,032 questions. It particularly emphasizes high-difficulty problems that require reasoning, and the number of answer choices has been expanded from 4 in the original MMLU to 10 to increase distractor complexity and discriminative power.

## C.3. Detailed Metrics

To rigorously evaluate the instruction following capabilities, we report the instruction-level accuracy defined by Eq. (2) in Table 1. Here we present detailed metrics of each benchmark introduced by its original paper. We find that HiR demonstrates consistent performance gains over baselines across different evaluation metrics.

**IFEval** adopts four metrics for evaluation: (1) *Prompt-level strict-accuracy* denotes strict criterion on the percentage of prompts that all verifiable constraints in each prompt are followed, (2) *Inst-level strict-accuracy* denotes strict criterion on the percentage of verifiable constraints that are followed, (3) *Prompt-level loose-accuracy* denotes loose criterion on the percentage of prompts that all verifiable constraints in each prompt are followed, and (4) *Inst-level loose-accuracy* denotes loose criterion on the percentage of verifiable constraints that are followed. Note that Prompt-level strict-accuracy is exactly the same as the ILA defined in our paper. Its detailed metrics are in Table 10.

*Table 10.* Detailed metrics of IFEval and IFBench. "Pr." and "Ins." denote prompt-level and instruction-level metrics respectively. "S" and "L" denote strict and loose respectively. **Bold** represents the best results among all methods.

| Model | IFEval | | | | IFBench | | | |
|---|---|---|---|---|---|---|---|---|
| | Pr. (S) | Pr. (L) | Ins. (S) | Ins. (L) | Pr. (S) | Pr. (L) | Ins. (S) | Ins. (L) |
| Llama-3.2-3B-Instruct | 71.2 | 75.8 | 78.5 | 82.7 | 23.8 | 26.9 | 25.4 | 30.1 |
| + SFT | 73.0 | 75.0 | 81.2 | 83.2 | 24.8 | 28.2 | 26.3 | 30.6 |
| + DPO | 74.3 | 78.6 | 82.4 | 85.9 | 22.1 | 27.6 | 24.5 | 29.9 |
| + RL-IR | 77.6 | 81.5 | 84.4 | 87.5 | 25.3 | 31.0 | 26.9 | 33.4 |
| + RL-CR | 79.1 | 82.4 | 85.6 | 87.9 | 26.6 | 29.6 | 28.6 | 33.1 |
| **+ HiR (Ours)** | **83.6** | **86.5** | **88.7** | **90.6** | **30.4** | **33.0** | **31.6** | **35.2** |
| Qwen2.5-7B-Instruct | 72.6 | 74.7 | 79.7 | 81.5 | 26.2 | 28.6 | 28.7 | 31.6 |
| + SFT | 75.6 | 79.1 | 82.0 | 84.8 | 27.9 | 33.3 | 29.3 | 35.5 |
| + DPO | 66.9 | 72.8 | 76.4 | 81.1 | 25.9 | 31.6 | 28.4 | 34.9 |
| + RL-IR | 76.2 | 79.5 | 82.8 | 85.4 | 31.1 | 36.7 | 34.0 | 40.6 |
| + RL-CR | 77.3 | 80.4 | 84.7 | 86.9 | 31.6 | 38.1 | 34.9 | 41.8 |
| **+ HiR (Ours)** | **81.0** | **83.2** | **86.7** | **88.4** | **35.8** | **41.5** | **38.2** | **44.7** |
| Qwen3-4B-Instruct-2507 | 83.4 | 87.4 | 89.1 | 91.4 | 29.9 | 33.0 | 31.9 | 35.2 |
| + SFT | 83.4 | 87.8 | 89.3 | 91.8 | 31.3 | 35.4 | 32.8 | 37.3 |
| + DPO | 83.9 | 88.0 | 89.6 | 92.1 | 27.9 | 31.0 | 29.9 | 34.0 |
| + RL-IR | 85.0 | 88.7 | 89.8 | 92.4 | 34.1 | 40.8 | 36.8 | 43.5 |
| + RL-CR | 85.8 | 88.2 | 90.4 | 92.1 | 36.9 | 42.9 | 38.8 | 47.2 |
| **+ HiR (Ours)** | **86.3** | **88.9** | **90.9** | **92.6** | **40.5** | **45.9** | **43.3** | **49.6** |

*Table 11.* Detailed metrics of CFBench, InfoBench, ComplexBench and FollowBench. **Bold** represents the best results among all methods.

| Model | CFBench | | | InfoBench | ComplexBench | FollowBench | | |
|---|---|---|---|---|---|---|---|---|
| | CSR | ISR | PSR | DRFR | DRFR | HSR | SSR | CSL |
| Llama-3.2-3B-Instruct | 67.2 | 31.3 | 42.6 | 78.5 | 62.1 | 58.0 | 69.8 | 2.4 |
| + SFT | 69.0 | 34.6 | 46.4 | 80.1 | 60.1 | 58.1 | 70.2 | 2.5 |
| + DPO | 73.0 | 40.1 | 52.6 | 79.9 | 62.9 | 61.8 | 72.1 | 2.6 |
| + RL-IR | 72.8 | 39.2 | 51.8 | 80.3 | 62.1 | 60.4 | 73.4 | 2.6 |
| + RL-CR | 72.6 | 38.9 | 51.3 | 80.0 | 62.7 | 61.1 | 73.9 | **2.7** |
| **+ HiR (Ours)** | **74.0** | **41.8** | **54.9** | **81.4** | **64.2** | **63.6** | **75.3** | **2.7** |
| Qwen2.5-7B-Instruct | 83.6 | 57.5 | 66.4 | 80.8 | 76.1 | 61.5 | 74.9 | 2.6 |
| + SFT | 82.1 | 53.1 | 65.4 | 80.7 | 74.9 | 62.6 | 75.8 | **2.8** |
| + DPO | 84.0 | 58.4 | 68.3 | 81.6 | 77.0 | **66.7** | **78.2** | **2.8** |
| + RL-IR | 85.2 | 60.1 | 70.0 | 82.1 | 78.2 | 62.3 | 76.1 | 2.7 |
| + RL-CR | 85.3 | 60.8 | 70.5 | 82.6 | 77.6 | 63.4 | 76.8 | **2.8** |
| **+ HiR (Ours)** | **86.8** | **64.2** | **72.8** | **84.5** | **79.8** | 65.1 | 77.3 | **2.8** |
| Qwen3-4B-Instruct-2507 | 88.1 | 67.5 | 76.2 | 85.2 | 80.0 | 76.1 | 84.0 | 3.3 |
| + SFT | 86.8 | 64.2 | 74.0 | 85.1 | 77.6 | 74.9 | 82.8 | 3.2 |
| + DPO | 88.7 | 68.0 | 76.9 | 84.9 | 80.3 | 78.0 | 85.6 | 3.4 |
| + RL-IR | 89.2 | 69.8 | 78.7 | 85.2 | 79.6 | 77.6 | 84.8 | 3.5 |
| + RL-CR | 88.9 | 68.5 | 77.3 | 85.4 | 81.0 | 78.2 | 85.7 | 3.5 |
| **+ HiR (Ours)** | **91.4** | **73.2** | **82.1** | **86.7** | **82.6** | **80.4** | **87.1** | **3.6** |

**IFBench** adopts the same metrics in IFEval for evaluation. Its detailed metrics are in Table 10.

**CFBench** adopts three metrics for evaluation: (1) *Constraint satisfaction rate (CSR)* denotes the average of satisfaction rate of each constraint across samples, (2) *Instruction satisfaction rate (ISR)* denotes the percentage of prompts that all constraints in each prompt are followed, and (3) *Priority satisfaction rate (PSR)* denotes the the percentage of prompts that priority constraints in each prompt are followed. Note that ISR is exactly the same as the ILA defined in our paper. Its detailed metrics are in Table 11.

**InfoBench** adopts *DRFR* metrics for evaluation, which denotes the average of satisfaction rate of each constraint across samples. Note that this is exactly the same as the constraint-level accuracy (CLA) defined in our paper. Its detailed metrics are in Table 11.

**ComplexBench** adopts the *DRFR* metrics in InfoBench for evaluation. Its detailed metrics are in Table 11.

**MulDimIF** calculates the percentage of prompts that all constraints in each prompt are followed. Note this is exactly the same as the instruction-level accuracy (ILA) defined in our paper, so we do not repeat report it here.

**FollowBench** adopts three metrics for evaluation: (1) *Hard Satisfaction Rate (HSR)* denotes the ILA defined in our paper, (2) *Soft Satisfaction Rate (SSR)* denotes the constraint-level accuracy (CLA) defined in our paper, and (3) *Consistent Satisfaction Levels (CSL)* denotes how many consecutive levels are satisfied per instruction. Its detailed metrics are in Table 11.

## D. Evaluation Prompt

We adopt the following prompt template for judging whether soft constraints are satisfied during RL training.

---

**Soft Constraints Evaluation Prompt Template**

Based on the provided Input (if any) and Generated Text, judge whether the generated text fulfills the Criteria Item with either a YES or NO choice. Your selection should be based on your judgment as well as the following rules:

- YES: Select 'YES' if the generated text entirely fulfills the condition specified in the Criteria Item. However, note that even minor inaccuracies exclude the text from receiving a 'YES' rating. As an illustration, consider a Criteria Item "Each sentence in the generated text uses a second person". If even one sentence does not use the second person, the answer should NOT be 'YES'. To qualify for a 'YES' rating, the generated text must be entirely accurate and satisfy the criteria.

- NO: Opt for 'NO' if the generated text fails to meet the criteria or provides no information that could be utilized to judge. For instance, the Criteria Item asks "Is the second sentence in the generated text a compound sentence?" and the generated text only has one sentence. It offers no relevant information to judge whether this criteria is met. Consequently, the answer should be 'NO'.

**Input:**
{input_text}
**Generated Text:**
{generated_text}
**Criteria Item:**
{criteria_item}

You only need to judge whether the generated text satisfy the given Criteria Item and do NOT affect by other requirements in Input (if any). Return either a 'YES' or 'NO' choice without any additional text in your response.

---

## E. Case Study

Here we provide detailed reasoning cases before and after training.

---

**Instructions from IFEval**

Write a poem about how I am missing my classes. The poem must have 4 sections marked with SECTION X. Finish the poem with this exact phrase: "Can I get my money back for the classes I missed?"

---

**Response of initial Qwen2.5-7B-Instruct**

SECTION X
In the quiet corners of my home,
Where shadows dance and whispers roam,
I miss the laughter, the shared jest,
The lessons that once filled my chest.

---

SECTION X
The echoes of the teacher's voice,
Now replaced by the hum of silence,
As I sit here, lost in my own choice.

SECTION X
The pages of my notebook, once so full,
Now empty, a silent, sad scroll,
The diagrams and equations, now mere memories,
Of a world that feels so far away, so free.

SECTION X
Can I get my money back for the classes I missed?
For the lessons that now seem so distant,
The friendships that now feel like a mist,
In the vast, lonely silence of this distant, silent distance.

**Response after trained by HiR**

**SECTION 1**
In the quiet corners of my home,
Where shadows dance and whispers roam,
I sit with books that once were mine,
But now they lie like silent signs.

**SECTION 2**
The chalk's sweet scent, a fleeting trace,
Of lessons learned, now far from grace.
The chalkboard's chalk, now just a dream,
Where knowledge once was, now just a scheme.

**SECTION 3**
The clock ticks on, a steady beat,
Counting down the hours, a lonely feat.
Each tick a reminder, sharp and clear,
Of the classes missed, a silent tear.

**SECTION 4**
The empty chair, a silent guard,
Of the lessons lost, a sad and hard.
The classroom's walls, now just a shell,
Of the dreams and hopes, now just a spell.

The chalk's last mark, a fading line,
Of the knowledge lost, now just a sign.
The teacher's words, a distant call,
For the classes missed, a silent fall.

Can I get my money back for the classes I missed?

In the following, we provide the examples of original instruction and its rewrited instruction in hindsight, with red denotes the unsatisfied constraints.

**Original Instruction**

Explain why some physicists argue that spacetime is fundamentally flawed and requires a radical rethinking, and describe the concept of a cosmological polytope in the context of reconciling quantum mechanics and general relativity.

**Requirements:**

- Restrict each sentence to 30 words maximum.

- Confine each paragraph to 2 sentences.

- The response should follow a clear and logical structure, with each paragraph building upon the previous one to create a cohesive narrative, and using transitional phrases to connect ideas and concepts.

- Employ metaphors and analogies to explain complex scientific concepts, such as comparing spacetime to a fabric or a network, and use vivid descriptions to illustrate the properties of cosmological polytopes.

- The response should be written in a formal and informative tone, conveying a sense of authority and expertise in the field of physics, while avoiding overly technical jargon and maintaining a level of accessibility for a general audience interested in scientific concepts.

- The main theme of the response should focus on the concept of spacetime and its potential flaws, with a clear explanation of the challenges posed by reconciling quantum mechanics and general relativity.

- Avoid using overly sensational language or provocative tone, and refrain from using colloquialisms or slang; instead, opt for precise and technical terminology when discussing complex scientific concepts.

**Rewrited Instruction**

Explain why some physicists argue that spacetime is fundamentally flawed and requires a radical rethinking, and describe the concept of a cosmological polytope in the context of reconciling quantum mechanics and general relativity.

**Requirements:**

- Restrict each sentence to 30 words maximum.

- The response should follow a clear and logical structure, with each paragraph building upon the previous one to create a cohesive narrative, and using transitional phrases to connect ideas and concepts.

- The response should be written in a formal and informative tone, conveying a sense of authority and expertise in the field of physics, while avoiding overly technical jargon and maintaining a level of accessibility for a general audience interested in scientific concepts.

- The main theme of the response should focus on the concept of spacetime and its potential flaws, with a clear explanation of the challenges posed by reconciling quantum mechanics and general relativity.

- Avoid using overly sensational language or provocative tone, and refrain from using colloquialisms or slang; instead, opt for precise and technical terminology when discussing complex scientific concepts.

**Response under Original Instruction**

Some physicists argue that spacetime may be fundamentally flawed because it cannot consistently reconcile the quantum mechanical behavior of particles with the continuous nature required by general relativity. This

incompatibility suggests that our current understanding of spacetime might need a radical overhaul.

General relativity predicts continuous, smooth spacetime, while quantum mechanics demands a discrete, granular structure. The transition between these regimes remains elusive, leading to conceptual and mathematical inconsistencies.

The concept of a cosmological polytope offers a promising approach to this challenge. They represent higher-dimensional shapes that can encapsulate the complex geometry of spacetime. By considering these polytopes, physicists aim to find a unified theory that integrates quantum mechanics and general relativity.

These polytopes could dynamically rearrange themselves under gravitational influences, reflecting both smooth and granular aspects. By doing so, they might provide a unified picture where quantum effects and gravitational forces coexist seamlessly, without the inherent contradictions found in current theories.

This approach suggests that the large-scale structure and evolution of the universe, described by general relativity, emerge from the interactions of these polytopes. At the same time, the quantum behavior of particles can be understood within this framework, offering a holistic view of the cosmos.

