# OpenReview forum: "Replay Failures as Successes: Sample-Efficient Reinforcement Learning for Instruction Following"
_ICML.cc/2026/Conference — ICML 2026 regular_

### Official Review · Reviewer_SXTD · 2026-02-27

**Soundness:** 3
**Presentation:** 4
**Significance:** 4
**Originality:** 3
**Overall Recommendation:** 4
**Confidence:** 4

**Summary:**

This paper proposes an RL-based approach to improve LLM’s instruction following ability under multiple constraints. The key challenge of this problem is that when the initial model is relatively weak, early-stage training will suffer from sparse or indistinguishable rewards. To address this, the authors propose HiR, a select-then-replay strategy that rewrites the prompt by removing unsatisfied constraints, thereby converting failure samples into successful samples and increasing reward density. Experimental results show that this method significantly improves performance across multiple popular instruction following benchmarks.

**Compliance With Llm Reviewing Policy:**

Affirmed.

**Final Justification:**

I am still somewhat uncertain about the use of entropy. But overall I am positive about this work, and the experiments are already very solid.

**Key Questions For Authors:**

See Weaknesses.

**Limitations:**

See Weaknesses.

**Strengths And Weaknesses:**

**Strengths**

- This method is conceptually clean. It improves reward signal density by rewriting the instruction to match the response, effectively converting failure rollouts into successful ones, without heavy cost.
- Empirically, this approach yields consistent and significant gains on a range of instruction following benchmarks, making the performance of smaller LLMs matching much strong LLM baselines.
- The paper provides solid and in-depth analysis. Sensitivity analysis and ablations help validate the effectiveness and robustness of the overall algorithm and each component. The additional analysis in Sec. 5 regarding training dynamics, parameter updates, and attention visualizations are informative and interesting, offering useful insights into why the method works.


**Weaknesses**
- In HiR, computing $F_{div}$ requires access to token-level probability distribution, which may introduce additional memory and runtime overhead. On the other hand, HiR reduces the number of samples per prompt. A quantitative breakdown of runtime and memory usage, together with comparisons to RL-IR/RL-CR would clarify the practical efficiency of the method.
- I’m not fully convinced that the design of $F_{div}$ captures the diversity we actually want. Intuitively, exploration should target diversity in constraint failure modes. Token-level entropy measures distributional uncertainty, which does not necessarily align with diversity over constraint violation patterns. It would strength the claim if the authors (i) include an ablation of $F_{div}$ and $F_{int}$ in Table 3, and (ii) report per-constraint success rates to demonstrate that $F_{div}$ indeed helps underrepresented or low-success constraints be better covered.
- The method seems to implicitly treat constraints as independent items that can be removed without affecting the semantics of the remaining instructions. It would be helpful to provide a brief discussion on the limitation when the constraints have dependencies, nested structures, or explicit priority.

---

> ### Author Rebuttal · Authors · 2026-03-31
>
> We sincerely thank you for your insightful comments and positive support!
>
> ---
>
> **[W1] A quantitative breakdown of runtime and memory usage.**
>
> Thanks for the suggestion! HiR requires fewer rollouts, which reduces both the sampling time and reward calculation time. The select process uses $F_{div}$ requires access to token-level probability distribution, but it will not introduce additional memory and runtime overhead. This is because RL itself requires the forward process to calculate `old_log_probs`, which contains token-level probability. It is a default operation and is usually integrated into mainstream RL infrastructures like verl. The rewrite process is a rule-based string modification (removing unmet constraints based on verifier feedback) that is computationally negligible compared to the gradient computation or the forward pass of the LLM. The subsequent training process is the same with standard RLVR. We provide detailed time in per RL step in Tab. R1. Since the number of samples is the same used for training, the memory usage is roughly the same compared to RL baselines. In summary, HiR does not increase per-step latency but reducing per-step latency, as shown in Fig. 7 of our manuscript.
>
> Table R1. Times per step (s) trained on Qwen3-4B-Instruct-2507
> |Time|Sampling|Reward|select-rewrite|Logprob|Update|Total|
> |-|-|-|-|-|-|-|
> |RL-CR|44.9|8.5|-|5.4|14.8|76.4|
> |HiR|35.7|6.4|0.9|3.5|14.6|64.6|
>
> **[W2] The disign of $F_{div}$**
>
> Thanks for the insightful comment! In this work, we adopt entropy that measures distribution-level diversity as $F_{div}$. This design is because low-probability tokens (distribution uncertainty) play an important role in RL training[1][2], which is more informative in reasoning. We agree that diversity in instruction-following could be also defined by constraint satisfication over a group of responses.
>
> (1) We have incorporated the ablation of $F_{int}$ and $F_{div}$ in Fig. 4 of our original manuscript. The results indicate emphasizing $F_{int}$ (large $\lambda_0$) may lead to insufficient exploration of the solution space; while prioritizing $F_{div}$ ($\lambda_0 = 0$) may fail to provide the necessary guidance required to satisfy all constraints in the later training stage. So both $F_{div}$ and $F_{int}$ are indispensable for optimal performance.
>
> (2) We further demonstrate that $F_{div}$ helps cover diverse constraints. Specifically, we define the constraint satisfaction vector $v_i \in \{0,1\}^n$ for each response $y_i$. Let $p(c_j)$ be the frequency of constraint $c_j$ is satisfied in the group $G$:$p(c_j)=\frac{\sum_{y \in G}v_{y,j}}{|G|}$. We define diversity $D(\tau)$ as the average entropy of the constraints satisfication probability: $D(\tau)=- \frac{1}{n} \sum_{j=1}^n [p(c_j) \log p(c_j)+(1-p(c_j)) \log (1-p(c_j))]$. Higher $D(\tau)$ indicates greater diversity in constraint satisfaction, with no concentration on specific categories of constraints. Let $T$ be the subset of responses with the top-k largest $F_{div}$ in the group, $S$ denotes the sum of the top-k largest values of $D(\tau)$ in the group. We measure coverage ratio as $\frac{\sum_{\tau_i \in T} D(\tau_i)}{S}$, which reflects how well the entropy-based selection aligns with the optimal constraint coverage. As shown in Tab. R2, selecting samples based on $F_{div}$ can achieve a high coverage ratio.
>
> Table R2.
> |Model|Ratio|
> |-|-|
> |Llama3.2-3B-Instruct|83%|
> |Qwen3-4B-Instruct-2507|86%|
>
> **[W3]  It would be helpful to provide a brief discussion on the limitation when the constraints have dependencies, nested structures, or explicit priority.**
>
> We greatly appreciate the reviewer for this profound and forward-looking comment! As a preliminary exploration of applying hindsight replay for LLMs, we simplify the potential relationships among constraints and treat them independently for convenience. This premise is consistent with popular instruction-following benchmarks like IFEval and IFBench, where constraints are explicitly designed as independent, verifiable atomic items. We acknowledge that HiR's current design has limitations such as possible logical contradiction when constraints have complex relationships. In future work, we can further extend HiR training data with structured constraint dependency to handle complex nested, dependent, and prioritized constraints. For example, we can introduce a lightweight constraint dependency graph parsing step in the rewrite phase, which only retains a constraint in the hindsight pseudo-instruction if all its prerequisite dependencies are satisfied. We will add a discussion of this limitation and future direction in our manuscript.
>
> ---
> **References**
>
> [1] Beyond the 80/20 Rule: High-Entropy Minority Tokens Drive Effective Reinforcement Learning for LLM Reasoning.
>
> [2] The Entropy Mechanism of Reinforcement Learning for Reasoning Language Models.

---

> > ### Author Rebuttal · Reviewer_SXTD · 2026-04-02
> >
> > Thank you for the clarifications. I will maintain my initial score, which was already positive.

---

> > > ### Author Response · Authors · 2026-04-03
> > >
> > > We sincerely thank the reviewer for the positive assessment of our work, including the conceptual clean method, good empirical results across benchmarks, and solid in-depth analysis. We are also grateful for the constructive comments, which improves the quality and clarity of our manuscript. Have a nice day :)

---

### Official Review · Reviewer_w44J · 2026-03-11

**Soundness:** 2
**Presentation:** 4
**Significance:** 2
**Originality:** 3
**Overall Recommendation:** 2
**Confidence:** 4

**Summary:**

This work proposes an algorithm to post-train large language models for instruction following where the instruction consists of multiple constraints. The main technique is to convert a negative response which may only satisfies partial constraints in the instruction to a successful response by re-constructing the instruction such that it only consists of constrains that being satisfied. By this way, they can replay the failure responses at each training step to train the model as if they are positive responses through an auxiliary loss on the original  GRPO loss. They also propose a selection strategy based on the diversity (entropy) and integrity (the number of constrains satisfied) of the response to select the top k replay samples with a curriculum which weighs integrity more as training proceeds. Finally, they conducted training on their curated datasets and evaluate on seven benchmarks along with some ablations to show the effectiveness of their method.

**Compliance With Llm Reviewing Policy:**

Affirmed.

**Final Justification:**

My final suggestion of this paper would be a reject. The theoretical issues which I listed in the weaknesses are not well resolved by the authors. The authors also admit that one should interpret the objective as a heuristic loss instead of a grounded one as claimed in the current version which is misleading. I feel the better presentation should be using the strictly correct importance-weighting ratio, and then use a tractable approximation. But I don't think it is just a presentation issue based on the current claim of contributions. At the same time, the experimental results look good. Overall, I suggest a reject.

**Key Questions For Authors:**

1. Which model are you using for LLM-as-a-judge? LLM-as-a-judge is just used for training on your own HIR-16K dataset, for evaluation benchmarks, there are still rule-based reward functions, is it correct?

2. What's limitations of your method? Based on your experiments, HiR is universally better than all other post-training paradigms, and does not affect the general reasoning ability? Did you do ablations on the training dataset? You trained the model on your own dataset, did you test your algorithm on other training datasets?

3. In equation (5), you define the diversity score with respect to one response by its entropy which is fine. However, it makes more sense to define the diversity score over a group of responses, for example, how many different constraints does this set of responses fulfill? Did you try selecting replayed samples with this group-based diversity metrics?

4. How is the advantage of replayed samples calculated? Isn't that for each different hindsight instructions, the number of replayed samples is limited? The reported k is 3, and the modified instructions are usually different since these responses may satisfy different subsets of the constraints, then it is very likely for a unique prompt, there is only one replayed sample.

5. Since the replayed samples for the hindsight instructions are all positive samples, why do you still use RL objectives, did you try just using supervised fine-tuning for the replayed samples?

**Limitations:**

As I asked in the questions section, it seems the proposed method is universally better than all other post-training paradigms for instruction following, I didn't see the author discussed the limitations of the proposed methods in the main page.

**Strengths And Weaknesses:**

1. There are two critical issues about the HiR objective in (8). First, since the replayed samples are selected/filtered, the importance ratio is incorrect. The probability of a response y being replayed is not $\pi_{old}(y)$. Second, the importance ratio ignores the state visitations, PPO (same for GRPO) are derived from performance difference lemma where it applies the approximation that $\pi(s)$ is similar to $\pi_{old}(s)$ where s denotes the state, in LLMs, it is the previous generated tokens. In the case where the prompt (instruction) is same, the approximation is fine, but here, the instructions are modified, this approximation no longer holds. Though one can regard replay objective in (8) as an auxiliary loss, it is not principled as the paper presents which may mislead readers. Also, I feel it makes more sense to use supervised-fine tuning for the replayed samples since they are all positive samples under the modified instructions.

2. The connection of HiR to preference learning in section 3.3 is weird and unnatural. In this way, any reinforcement learning algorithm can be formulated as preference learning where a win response is with positive advantages and a lose response is with negative advantages which makes no sense. By the way, there is a typo in Proposition 3.2, it should be $\nabla J_{HiR}(\theta)$. I feel this connection is unnecessary. It is not truly preference learning, and building this connection does not bring any insights on the theoretical understanding nor the practical algorithm designs.

3. The authors did extensive experiments and present the results in a very clean and easy-to-read way. The experimental results are surprisingly good. HiR shows universally better performance on all seven benchmarks. One question is HiR should show better advantages  on hard benchmarks, since even though all responses are negative, HiR still imposes positive learning signals, but I didn't see this trend in the experiments section.

4. The main idea is inspired from the goal-conditional reinforcement learning, where a failure trajectory can be regarded as success by altering the goal. It is interesting to apply this idea to LLMs.

---

> ### Author Rebuttal · Authors · 2026-03-31
>
> Tables in https://anonymous.4open.science/r/HIR_Code-ECF2/re.md
>
> **[W1&Q5] HiR objective**
>
> Thanks for important observation and suggestion!
>
> (1) The ratio in Eq. (8) can be interpreted as a PPO-inspired weighting mechanism. Eq. (9) can be viewed as decomposing the weighting as $\rho_t=r_t \cdot k_t$, where $r_t = \frac{\pi(y_t|q', y_{<t})}{\pi_{old}(y_t|q', y_{<t})}$ and $k_t = \frac{\pi_{old}(y_t|q', y_{<t})}{\pi_{old}(y_t|q, y_{<t})}$. Here, goal-conditioned IS $k_t$ reflects the change in action likelihood under the modified instruction q' relative to q, serving as a heuristic adjustment for instruction shift. Moreover, $q'$ is obtained via minimal edits to $q$, the semantic shift is limited, which empirically makes replay effective. We will clarify this in the revision.
>
> (2) We provide additional results that uses supervised fine-tuning on replayed pairs, jointly trained with the RL objective on original instructions. Tab. R1 shows that HiR under pure RL framework has better robustness and generalization ability compared to SFT hybrid training.
>
> **[W2] The connection of HiR to preference learning**
>
> Thanks for pointing out typos! We clarify that this connection serves primarily as an analytical lens to interpret the mechanism of HiR instead of a design principle for algorithm. In fact, standard RLHF optimizes the generated response (under a given instruction) to maximize alignment with human preferences. HiR induces a dual preference structure: it not only learns preference on different responses but also motivates a deeper investigation into the preference of instructions relative to a response. We will revise to make this distinction clearer and avoid overclaiming.
>
> **[W3] HiR should show better advantages on hard benchmarks**
>
> As analyzed in Sec. 4.2, when the capability of the model increases, performance gains on saturated metrics like IFEval diminish, yet advantages remain pronounced on more challenging datasets, such as IFBench and MultiDimIF.
>
> **[Q1] Judge model**
>
> DeepSeek-V3.1 are used to judge in both training and evluation, as stated in 4.1. For benchmarks like ComplexBench have both soft and hard constraints in instructions. So they need both rule verifier and llm to score.
>
> **[Q2] Limitations and ablations on the training dataset**
>
> Thanks! We have discussed our limitations in Appendix A.3. Besides, as a preliminary exploration, we do not incorporate constraints with complex dependencies in training. We further trained on IFTrain[1] to validate the effectiveness of HiR. As shown in Tab. R2, it performs well on other training sets.
>
> **[Q3] Selecting replayed samples with group-based diversity metrics**
>
> Thanks for the insightful suggestion! In this work, we adopt entropy that measures distribution-level diversity as $F_{div}$. This design is because low-probability tokens (distribution uncertainty) play an important role in RL training, which is more informative in reasoning. We agree that diversity in instruction-following could be also defined by constraint satisfication over a group of responses.
>
> We define the constraint satisfaction vector $v_i \in \{0,1\}^n$ for each response $y_i$. Let $p(c_j)$ be the frequency of constraint $c_j$ is satisfied in the group $G$:$p(c_j)=\frac{\sum_{y \in G}v_{y,j}}{|G|}$. We define diversity $D(\tau)$ as the average entropy of the constraints satisfication probability: $D(\tau)=- \frac{1}{n} \sum_{j=1}^n [p(c_j) \log p(c_j)+(1-p(c_j)) \log (1-p(c_j))]$. Higher $D(\tau)$ indicates greater diversity in constraint satisfaction, with no concentration on specific categories of constraints. We replace $F_{div}$ with group-based metrics $D(\tau)$ to conduct experiments. As shown in Tab. R3, this method did not demonstrate superior performance compared to the original method, indicating diversity in token distribution may be better for RL learning.
>
> **[Q4] How is the advantage of replayed samples calculated**
>
> Sorry for the confusion. First, the computation of return $G_{ij}(s_t, y_t)$ at state $s_t$ is $G_{ij}(s_t,y_t)=\sum_{k=t}^T R_{ij}(s_k, y_k)-KL_{ij}(k)$, where $R_{ij}(s_k, y_k)$ denotes the token-level reward at the $k$-th token of the $j$-th response to the $i$-th question and $KL_{ij}(k)$ denotes the token-level KL divergence at the $k$-th token between the current model and the initial model. Based on the return $G(s_t,y_t)$, assuming a training batch contains $n_b$ questions and each question is associated with $n_g$ responses, we compute the mean and standard deviation of the return across the batch as $$mean_t = \frac{1}{n_b \cdot n_g} \sum_{i=1}^{n_b} \sum_{j=1}^{n_g} G_{ij}(s_t, y_t), std_t = \sqrt{\frac{1}{n_b \cdot n_g} \sum_{i=1}^{n_b} \sum_{j=1}^{n_g} \left( G_{ij}(s_t, y_t) - mean_t \right)^2}$$. Then, the advantage is computed as $A_{ij}(s_t, y_t)=\frac{G_{ij}(s_t, y_t)-mean_t}{std_t}$. To summary, it's unrelated to the number of replay samples under modified instructions.
>
> ---
> [1] Generalizing Verifiable Instruction Following.

---

> > ### Author Rebuttal · Reviewer_w44J · 2026-04-04
> >
> > Thank the authors for the efforts on the rebuttal. However, my main concern is not resolved yet.
> >
> > (1). You sampled responses from a policy, let's say the old policy. Without doing any rejection sampling, then your importance weighting ratio is correct, but since you are doing rejection sampling for the replayed responses (which is using scores in (5) and (6) to select top-k), the probability that a response will be selected is no longer the original probability. For example, you roll a dice, and get a dataset of rollouts, then sample number from this dataset, yes, the probability of each number being sampled is 1/6. But if you select multiple numbers from this dataset and just take the biggest one, then the probability of each number being selected is no longer 1/6, but you are still using 1/6 as the denominator for the importance weighting ratio which is incorrect.
> >
> > (2). Another concern, the explanation on q' and q are semantically similar to support that the state visitation does not deviate much is weak.
> >
> > (3). Finally, the way you calculate the advantages for the replayed samples are weird. You are calculating advantages using the mean reward of all rollouts of different prompts. Though, in theory, there is no problem, any action-independent value can be used as a baseline for policy gradient methods, it should cause problems in practice. Usually, in standard GRPO, we calculate advantages with respect to the mean reward of the rollouts for the same prompt, which captures how good a rollout is compared to the expected performance of the current policy. And this interpretation no longer holds if you calculate mean reward using rollouts for all prompts.
> >
> > Based on the above remaining concerns, I don't plan to change the score currently.

---

> > > ### Author Response · Authors · 2026-04-07
> > >
> > > We sincerely appreciate the reviewer for constructive feedbacks. Below we further clarify the remaining points.
> > >
> > > **[W1] Importance ratio.**
> > >
> > > (1) We note that the selection of replayed samples changes the data distribution $\pi_{old}(y)$, and therefore the importance ratio should not be interpreted as an unbiased correction but a PPO-inspired weighting mechanism. The selection process plays a role in focusing optimizations on informative failures, which improves learning efficiency. In practice, such selection is widely adopted for LLMs in the community [1,2], where only successful samples are filtered for replay to accelerate convergence and stabilize training despite deviating the original probability in importance ratio. Although strict importance correction is typically not enforced, its empirical effectiveness is well established. We will revise our manuscript to explicitly clarify this design.
> > >
> > > (2) To further investigate this concern, we modify our original objective to remove selection bias. Specifically, we select $k=2$ samples for replay according to the probability in a sampling group. The probability of response $y_i$ being selected is $P(y_i)=\frac{F(y_i)}{\sum_{y\in\mathcal{G}} F(y)}$, where $F(y)$ denotes the score of response $y$ based on diversity and integrity. We additionally introduce an importance sampling weights $w_i=\frac{1}{m}\cdot\frac{1}{P(y_i)}$ in our objective to correct the selection bias, which preserves the expected gradient mathematically:
> > >
> > > $$
> > > \mathcal{J}\_{\text{HiR-Weighted}}(\theta)=\mathbb{E}\_{q \sim \mathcal{D},\{y^{(i)}\}\_{i=1}^{m} \sim \pi\_{\text{old}}(\cdot|q),\{q'^{(i)},y'^{(i)}\}\_{i=1}^{k} \sim \mathcal{H}} \bigg[\underbrace{\frac{1}{m} \sum\_{i=1}^{m} \frac{1}{|y^{(i)}|} \sum\_{t=1}^{|y^{(i)}|} \min \left(\rho\_{t, \theta}^{(i)} A_{t}^{(i)},\text{clip}(\rho\_{t, \theta}^{(i)},1 \pm \epsilon) A\_{t}^{(i)} \right)}\_{\text{Objective for Initial Samples}}+\underbrace{{\frac{1}{k} \sum\_{i=1}^k \frac{\textcolor{red}{w'^{(i)}}}{|y'^{(i)}|} \sum_{t=1}^{|y'^{(i)}|} \min \left(\rho\_{t,\theta}'^{(i)} A\_{t}'^{(i)},\text{clip}(\rho\_{t,\theta}'^{(i)},1 \pm \epsilon) A\_{t}'^{(i)} \right)}}\_{\text{Objective for Replayed Samples}}\bigg].
> > > $$
> > >
> > > As shown in Tab. R5, although theoretically unbiased, it does not demonstrate superior results compared to the current objective. Moreover, correcting selection bias introduces higher variance in gradient, which may offset its theoretical advantages. This indicates focusing solely on selected informative failures for replay can be effective.
> > >
> > > Table R5. Model: Qwen3-4B-Instruct-2507.
> > > |Objective|IFEval|IFBench|CFBench|InfoBench|ComplexBench|MulDimIF|FollowBench|
> > > |-|-|-|-|-|-|-|-|
> > > |Original HiR|86.3|40.5|73.2|60.8|61.5|80.6|80.4|
> > > |HiR+Weighted IS|86.3|40.1|73.0|61.4|61.9|80.1|80.8|
> > >
> > > **[W2] State probability under q and q'.**
> > >
> > > (1) First, we employ goal-conditioned IS $\frac{\pi_{old}(y_t|q',y_{<t})}{\pi_{old}(y_t|q,y_{<t})}$ in our importance ratio (Eq. 9) to reflect the change in state likelihood under the rewritten instruction q' relative to q, serving as a heuristic adjustment for potential "instruction shift". Besides, the clip mechanism also restricts the update of tokens with large deviations.
> > >
> > > (2) Moreover, q' is derived by minimally removing unmet constraints in q, thus the semantic shift between them is limited. We further provide empirical results on the deviation of state probability under q and q' in https://anonymous.4open.science/r/HIR_Code-ECF2/re.md. As shown in Tab. R4, Fig. 1 and Fig. 2 in the url, the state distribution is almost identical with limited distribution shift between original instruction q and rewritten instruction q'. Quantitatively, $\frac{1}{|y|}\sum_{t=1}^{y}\frac{\pi_{old}(y_t|q',y_{<t})}{\pi_{old}(y_t|q,y_{<t})}=0.9924$, which are approximately equal.
> > >
> > > **[W3] Advantage Calculation.**
> > >
> > > We would like to clarify that our advantage calculation is the default design of Reinforce++ [3] algorithm. The advantage estimation in Reinforce++ uses **batch-level normalization** of all rollouts in a batch (with different prompts), which is different from GRPO that calculates advantage using **group-level normalization** of the rollouts in a sampling group. Many works [4,5] in the community have validated that Reinforce++ with batch-level normalization can yield stable training dynamics and better performance compared to GRPO.
> > >
> > > Thanks again for the time and insightful comments on our manuscript and rebuttal. We hope our response will address your concerns, and we genuinely look forward to further discussion and re-evaluation of our work.
> > >
> > > ---
> > > **References**
> > >
> > > [1] RLEP: Reinforcement Learning with Experience Replay for LLM Reasoning.
> > >
> > > [2] ExGRPO: Learning to Reason from Experience. ICLR 2026
> > >
> > > [3] REINFORCE++: Stabilizing Critic-Free Policy Optimization with Global Advantage Normalization.
> > >
> > > [4] Logic-RL: Unleashing LLM Reasoning with Rule-Based Reinforcement Learning.
> > >
> > > [5] Process Reinforcement through Implicit Rewards.

---

### Official Review · Reviewer_tm5g · 2026-03-11

**Soundness:** 3
**Presentation:** 3
**Significance:** 2
**Originality:** 2
**Overall Recommendation:** 4
**Confidence:** 3

**Summary:**

This paper proposes Hindsight Instruction Replay (HiR), where the author proposes a RL algorithm with replay buffer to provide more informative signals during training. The author frames the training pipeline with select-then-replay pipeline with curriculum selection strategies, where initially diversity is focused but then changed to constraint integrity. The method outperforms baseline on many tasks.

**Compliance With Llm Reviewing Policy:**

Affirmed.

**Final Justification:**

Thanks for the reply. Many of my concerns are resolved. I will raise my score accordingly. Nevertheless, more thorough comparisons over dense reward training algorithms, such as treerpo, and more ablations on powerful models are encouraged.

**Key Questions For Authors:**

Why is the pseudo code of the main algorithm not given in the main paper? I would suggest to have a better illustration of the algorithm in the main paper since appendix is optional.

**Limitations:**

See my weakness. My main concern is the overhead analysis and deeper dive of the key novelty compared to other replay buffer papers.

**Strengths And Weaknesses:**

Strengths:

1.	The empirical results look good on different benchmarks and the analysis with the training dynamics is sufficient

2.	The author has many ablation studies to study this paper.

3. The motivation and the methdology is clearly illustrated.


Weakness

1.	Unclear key novelties. The paper focuses on utilizing replay buffer to facilitate the training. I didn’t see sufficient discussions compared to other related papers, see [1,2,3]. More discussions with the differences and the strengths over these methods should be included.

2.	Comparison to other methods, like self-refinement and self-critic is also limited,

3.	Though the author claims it’s more efficient in time, but I only see steps included in the figure. I think the author’s method would add great overhead regarding to the per-step latency. However, this is not analyzed in the paper.

4. Would the assumption of the insufficient learning signal for not being able to solve the problems well during initial stages still hold true for more capanle models and better training algorithms (like gspo)?

[1]: RePO: Replay-Enhanced Policy Optimization
[2]: RLEP: Reinforcement Learning with Experience Replay for LLM Reasoning
[3]: ExGRPO: Learning to Reason from Experience

---

> ### Author Rebuttal · Authors · 2026-03-31
>
> We sincerely thanks for your time and insightful comments on our submission.
>
> **[W1] Unclear key novelties.**
>
> Thanks for the suggestion! We clarify that our key novelty is not introducing a replay buffer, but focusing on **how to convert failure attempts into successful ones through select-then-rewrite strategy**. Unlike RePO/RLEP/ExGRPO, which maintains a historical replay buffer and reuses historical sampled trajectories under the original instruction to improve data efficiency, HiR does not store and reuse past trajectories from previous training steps. RePO/RLEP/ExGRPO improve sample utilization of previous trajectories, but cannot resolve reward sparsity or reward ambiguity in multi-constraint tasks. Instead, HiR relabels failed trajectories by rewriting the original instruction into a hindsight pseudo-instruction containing only the satisfied constraints, thereby converting failures into positive training samples that are unavailable in ordinary replay buffers. By this way, HiR enriches the learning signal in training stage. To the best of our knowledge, we are the first to apply this idea to LLMs. We will add the discussions on differences with these methods to our manuscript.
>
>
> **[W2] Comparison to self-refinement and self-critic is limited.**
>
> Thanks! Self-refinement/critic rely on multiple rounds of iterative generation, while HiR is a training framework that converts failed responses into successful ones during training. These two methods do not overlap much with HiR, so we did not compare them. To address your concerns, we design experiments to incorporate self-refinement in RL sampling for comparison. Specifically, we sample 6 original responses and using the unmet constraints as guidance to generate 2 successful refinements. Both refinements and original responses are used for RL training. As shown in Table R1, HiR still show better performance.
>
> Table R1. Model: Qwen3-4B-Instruct-2507
> |Method|IFEval|IFBench|MulDimIF|
> |-|-|-|-|
> |Self-refinement + RL|86.1|38.8|80.2|
> |HiR|86.3|40.5|80.6|
>
> **[W3] I think the author’s method would add great overhead.**
>
> Thanks! We clarify that HiR does not add overhead regarding to the per-step latency compared to other RL baselines. In fact, the design of HiR is lightweight and time-efficient. In RL settings, we ensure consistent numbers of training data (8 per group) for fair comparison. Specifically, HiR generates only 6 responses and replays 2 (total 8 samples per group), whereas baseline RL-IR/RL-CR require 8 rollouts. Therefore, HiR requires fewer rollouts, which reduces both the sampling time and reward calculation time. The select-then-rewrite process in HiR is a rule-based instruction modification (removing unmet constraints based on verifier feedback). This is a string operation that is computationally negligible compared to the gradient computation or the forward pass of the LLM. We further provide detailed time in per RL step in Table R2. In summary, HiR does not increase per-step latency but reducing per-step latency, as shown in Figure 7 of our manuscript.
>
> Table R2. Times per step (s) trained on Qwen3-4B-Instruct-2507
> |Time|Sampling|Reward|select-rewrite|Logprob|Update|Total|
> |-|-|-|-|-|-|-|
> |RL-CR|44.9|8.5|-|5.4|14.8|76.4|
> |HiR|35.7|6.4|0.9|3.5|14.6|64.6|
>
> **[W4] Would the assumption hold true for more capanle models and better training algorithms?**
>
> Thanks for the question! Our motivation holds for stronger models and better algorithms.
>
> (1) Stronger models have to learn on more difficult problems. Recent studies [1][2] have revealed that focusing on more difficult problems can better enhance capability of LLMs. Therefore, the issue of insufficient positive signals exists.
>
> (2) Advanced RL algorithms like GSPO optimize the policy gradient estimation and update rules to improve training stability, but they cannot solve the fundamental problem of insufficient positive learning signals when successful samples are scarce.
>
> **[Q] Why is the pseudo code not given in the main paper?**
>
> Thanks. Due to page limit of submission, we put it in appendix. We'll move to main paper in revisions.
>
> ---
> [1] Imitate, Explore, and Self-Improve: A Reproduction Report on Slow-thinking Reasoning Systems.
>
> [2] Kimi k1.5: Scaling Reinforcement Learning with LLMs

---

> > ### Author Rebuttal · Reviewer_tm5g · 2026-04-04
> >
> > Thanks for the reply. Many of my concerns are resolved. I will raise my score accordingly. Nevertheless, more thorough comparisons over dense reward training algorithms, such as treerpo, and more ablations on powerful models are encouraged.

---

> > > ### Author Response · Authors · 2026-04-07
> > >
> > > We greatly appreciate the reviewer for the time in reviewing our rebuttal and positive re-evaluation. As suggested, we employ the idea of TreeRPO [1] that produces dense reward in training. The additional results are provided in Table R3. We will incorporate these results in our revisions.
> > >
> > > Table R3. Model: Qwen3-4B-Instruct-2507.
> > > | Method  | IFEval   | IFBench  | MulDimIF |
> > > | ------- | -------- | -------- | -------- |
> > > | TreeRPO | 86.1     | 37.8 | 79.8 |
> > > | HiR     | 86.3 | 40.5     | 80.6     |
> > >
> > > ---
> > >
> > > **References**
> > >
> > > [1] TreeRPO: Tree Relative Policy Optimization.

---

### Official Review · Reviewer_Z9J9 · 2026-03-13

**Soundness:** 3
**Presentation:** 3
**Significance:** 2
**Originality:** 2
**Overall Recommendation:** 4
**Confidence:** 3

**Summary:**

This paper proposes Hindsight instruction Replay (HiR) that revises failed instructions and replays the revised instruction-response pairs to improve the performance of large language models (LLMs) for instruction following. This paper evaluates HiR across different instruction following benchmarks (e.g., IFEval, IFBench, CFBench, etc.) and different LLMs (e.g., Llama-3.2-3B-Instruct, Qwen2.5-7B-Instruct, and Qwen3-4B-Instruct-2507). Evaluation results show that HiR can consistently improve the performance of LLMs across different instruction following benchmarks.

**Compliance With Llm Reviewing Policy:**

Affirmed.

**Final Justification:**

This paper proposes Hindsight instruction Replay (HiR) that replays the revised instruction-response pairs in order to improve the instruction following ability of large language models (LLMs). In my review, I raised one weak point (i.e., the limited rewriting strategy) and one question (i.e., the applicability of HiR to other RL methods). The responses provided by the authors largely resolved my concerns and questions. Therefore, I maintain my initial positive score.

**Key Questions For Authors:**

- Q1. In this paper, HiR is mainly applied to REINFORCE++. Can HiR be used with other RL methods such GRPO?

**Limitations:**

yes

**Strengths And Weaknesses:**

Some strengths of this paper can be summarized as follows:
- S1. First of all, this paper is clearly written and well organized.
- S2. This paper proposes the select-then-rewrite replay strategy for hindsight instruction replay. It seems novel and practical. Also, this paper provides a theoretical analysis on the method.
- S3. This paper provides comprehensive experiment results by evaluating HiR across different instruction following benchmarks (e.g., IFEval, IFBench, CFBench, etc.) and different LLMs (e.g., Llama-3.2-3B-Instruct, Qwen2.5-7B-Instruct, and Qwen3-4B-Instruct-2507).
- S4. The performance improvements of HiR are impressive, especially on IFBench and MulDimIF.

Some weaknesses of this paper can be summarized as follows:
- W1. The way of rewriting failed instructions is mainly to remove the unmet constraints. This rewriting method seems rather limited. Can HiR include other rewriting methods such as rewriting the instruction itself?

---

> ### Author Rebuttal · Authors · 2026-03-31
>
> We sincerely thank you for your insightful review and positive support!
>
> ---
>
> **[W1] This rewriting method seems rather limited.**
>
> Thanks for the insightful comments! Our HiR framework is compatible with other rewriting methods since the rewriting module is decoupled from other parts. The method can be replaced with any valid hindsight rewriting strategy, as long as the original response satisfies all constraints in the rewritten pseudo-instruction. Beyond constraint removal, we can refine the unmet constraint to match the response using LLMs, rather than removing it entirely. For example, if the original instruction contains "write a poem about spring" and the response is a poem about summer, we can rewrite the instruction to "write a poem about summer" using LLMs, which retains more constraint information while maintaining the validity of the positive sample after hindsight rewriting.
>
> In this work, we adopt a deterministic and lightweight hindsight rewriting solution just by removing the unmet constraints, which avoids additional rewriting noise or hallucinated constraints introduced by LLMs. This provides a simple yet stable way to validate the effectiveness of our proposed HiR. We agree that more flexible rewriting methods, such as paraphrasing or re-expressing the constraints in instruction, are promising directions for future work.
>
> **[Q1] Can HiR be used with other RL methods?**
>
> Thanks for your question! HiR is a hindsight replay framework for LLMs that is decoupled from the underlying RL algorithm, and can be integrated with different RL algorithms. The core of HiR is to convert failed attempts into successful samples through the select-then-rewrite strategy to enrich the the positive learning signals for subsequent policy optimization, while not changing the underlying policy optimization principle. As shown in Table R1, HiR can optimize using PPO algorithm (Note that GRPO, Reinforce++, RLOO, etc. are simplifications or variations of PPO).
>
> Table R1. Model: Qwen3-4B-Instruct-2507.
> |Algorithm|IFEval|IFBench|CFBench|InfoBench|ComplexBench|MulDimIF|FollowBench|
> |--|--|--|--|--|--|--|--|
> |HiR (Reinforce++)|86.3|**40.5**|73.2|60.8|61.5|80.6|**80.4**|
> |HiR (PPO)|**86.7**|39.5|**74.4**|**61.2**|**61.9**|**82.3**|80.1|

---

> > ### Author Rebuttal · Reviewer_Z9J9 · 2026-04-04
> >
> > I thank the authors for providing thoughtful responses to my comments and questions. The answers provided by the authors largely resolved my questions. Especially, the experiment result on PPO is interesting to me, and this shows that HiR can be integrated with different RL algorithms. Therefore, I maintain my initial positive score.

---

> > > ### Author Response · Authors · 2026-04-07
> > >
> > > Thanks for reviewing our rebuttal and the positive assessment of our work! Your initial questions inspired us to dive deeper into the rewriting mechanism and underlying optimization algorithms, which strengthens the manuscript. We greatly appreciate your time and insightful feedback, and will include the discussion and results in revisions.

---

### Decision · Program_Chairs · 2026-04-30

**Decision:**

Accept (regular)

**Comment:**

The paper proposes Hindsight Instruction Replay (HiR, inspired by HER in traditional deep RL) to mitigate sparse rewards in instruction following by rewriting instructions to match the constraints actually satisfied by the model's responses. Reviewers gave the work scores of 4, 4, 4, and 2, broadly highlighting the "universally better" performance across diverse benchmarks and the framework's computational efficiency. The most critical feedback concerned the theoretical groundedness of the importance sampling ratio; however, the authors effectively argued that their approach is a practical PPO-inspired weighting mechanism common in LLM alignment. They successfully showed that mathematically "correcting" this bias actually degrades performance by increasing variance. Given the overwhelming empirical evidence and the clear utility of the hindsight mechanism for training weaker models on complex tasks, the AC recommends acceptance.